
# Multi-centered black holes, scaling solutions and pure-Higgs indices from localization

Guillaume Beaujard[1*], Swapnamay Mondal[2,3†] and Boris Pioline[1‡]

**1** Laboratoire de Physique Théorique et Hautes Energies (LPTHE), UMR 7589
CNRS-Sorbonne Université, Campus Pierre et Marie Curie,
4 place Jussieu, F-75005 Paris, France
**2** School of Mathematics, Trinity College, Dublin 2, Ireland
**3** Hamilton Mathematical Institute, Trinity College, Dublin 2, Ireland

* beaujard@lpthe.jussieu.fr, † swapno@maths.tcd.ie, ‡ pioline@lpthe.jussieu.fr

## Abstract

The Coulomb Branch Formula conjecturally expresses the refined Witten index for $\mathcal{N} = 4$ Quiver Quantum Mechanics as a sum over multi-centered collinear black hole solutions, weighted by so-called 'single-centered' or 'pure-Higgs' indices, and suitably modified when the quiver has oriented cycles. On the other hand, localization expresses the same index as an integral over the complexified Cartan torus and auxiliary fields, which by Stokes' theorem leads to the famous Jeffrey-Kirwan residue formula. Here, by evaluating the same integral using steepest descent methods, we show the index is in fact given by a sum over deformed multi-centered collinear solutions, which encompasses both regular and scaling collinear solutions. As a result, we confirm the Coulomb Branch Formula for Abelian quivers in the presence of oriented cycles, and identify the origin of the pure-Higgs and minimal modification terms as coming from collinear scaling solutions. For cyclic Abelian quivers, we observe that part of the scaling contributions reproduce the stacky invariants for trivial stability, a mathematically well-defined notion whose physics significance had remained obscure.

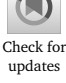

# 1 Introduction

For the purpose of determining the BPS spectrum in supersymmetric field theories or string vacua with $\mathcal{N} = 2$ supersymmetry in four dimensions, a special class of supersymmetric quantum mechanics known as $\mathcal{N} = 4$ Quiver Quantum Mechanics (QQM) provides an essential tool. Indeed, it describes the low energy dynamics of a set of $K$ dyons with general, mutually non-local electromagnetic charges $\gamma_i$, each of them separately saturating the BPS bound, and interacting through the usual Coulomb, Lorentz, scalar exchange and (when coupled to gravity) Newton forces. When the dyons have distinct charges $\gamma_i$, QQM is an 0+1 dimensional $U(1)^K$ gauge theory with charged matter determined by the Dirac-Schwinger-Zwanziger pairing $\kappa_{ij} = \langle \gamma_i, \gamma_j \rangle$ between the charges of the constituents, encoded in a quiver $Q$. More generally, it is a non-Abelian gauge theory with gauge group $\prod_{i=1}^{K} U(N_i)$ and bifundamental matter, whose Lagrangian follows from the usual rules of string theory whenever the dyons can be viewed as wrapped D-branes [1, 2]. Semi-classically, its moduli space consists of a Coulomb branch, where the gauge group is broken to the Cartan torus and the scalars $\vec{x}_i$ in the vector multiplet satisfy the same equations as multi-centered BPS black holes in $\mathcal{N} = 2$

supergravity [2,3],

$$\forall i, \quad \sum_{j \neq i} \frac{\kappa_{ij}}{|\vec{x}_i - \vec{x}_j|} = 2\zeta_i , \tag{1.1}$$

and a Higgs branch where the $\vec{x}_i$'s are coincident and non-Abelian degrees of freedom become important. Both branches carry an action of the R-symmetry subgroup $SO(3)_R$, which corresponds to rotations in spatial directions.

While the quantum dynamics of QQM is complicated, the refined Witten index $\Omega(\gamma, y, \zeta)$ counting supersymmetric ground states weighted by their angular momentum $y^{2J_3}$ (and therefore BPS bound states of the $K$ dyons with total charge $\gamma = \sum_i \gamma_i$), can be evaluated using localization [4] (see also [5–7]). Crucially, the index depends on the Fayet-Iliopoulos parameters $\zeta$ and superpotential $W$, with jumps in real codimension one or two, respectively, when these data are varied. The former corresponds to the familiar wall-crossing phenomena in four-dimensional theories with $\mathcal{N} = 2$ supersymmetry (see e.g. [8] and references therein), while the latter will be irrelevant in this work, where we shall assume $W$ to be generic.

Mathematically, the refined Witten index (also known as refined Donaldson-Thomas invariant) is given by the $\chi_{y^2}$-genus of the moduli space $\mathcal{M}_Q(\gamma, \zeta)$ of stable quiver representations with dimension vector $\gamma = (N_1, \dots, N_K)$. The localization computation of [4] (closely related to the elliptic genus computation in two-dimensional gauged linear sigma models [9]) expresses $\Omega(\gamma, y, \zeta)$ as a Jeffrey-Kirwan residue formula, with a contour prescription depending on the stability parameters $\zeta$. For generic values of $\zeta$ away from the walls, $\Omega(\gamma, y, \zeta)$ evaluates to a symmetric Laurent polynomial in $y$, corresponding to the character of the action of the rotation $y^{2J_3}$ on BPS ground states, and reduces to the usual Witten index $\Omega(\gamma, \zeta)$ as $y \to 1$, equal (up to sign) to the Euler number of the moduli space of stable quiver representations.

In a series of works by J. Manschot, A. Sen and the third named author [10–14], an alternative, heuristic description of the supersymmetric ground states of QQM was proposed, by localizing the effective supersymmetric quantum mechanics on the Coulomb branch to fixed points of the rotation $J_3$. Not surprisingly, fixed points of $J_3$ are collinear configurations of $K$ dyons localized along the $z$ axis, whose relative distances are fixed by a one-dimensional version of (1.1),

$$\forall i, \quad \sum_{j \neq i} \frac{\kappa_{ij}}{|z_i - z_j|} = 2\zeta_i . \tag{1.2}$$

Summing over all possible orderings with a suitable sign, and assigning unit degeneracy $\Omega(\gamma_i, y) = 1$ to each constituent, this prescription produces the correct refined Witten index for Abelian quivers without oriented cycles. This prescription also extends to non-Abelian quivers without oriented cycles, provided the dyons are treated as Boltzmannian particles (i.e. distinguishable) and weighted by an effective rational index $\bar{\Omega}(\gamma_i, y)$. In contrast, when the quiver has oriented cycles, this prescription fails to produce a bona-fide character of the rotation group, rather it produces a rational function of $y$ with a pole at $y = 1$. This issue can be traced to the existence of fixed points of $J_3$ which do not correspond to any collinear configuration, but rather to 'scaling solutions', where the centers become arbitrarily close to each other, with almost vanishing angular momentum [15–17].

In [11], an *ad hoc* prescription was proposed to rectify this problem, by modifying the naive count of collinear solutions and introducing new, so called 'single-centered indices' $\Omega_S(\gamma, y)$ (also known as 'pure-Higgs' or 'intrinsic Higgs') counting pointlike configurations with total charge $\gamma = \sum_i N_i \gamma_i$, whenever the dimension vector $\gamma$ is supported on a subquiver which contains an oriented cycle. Unlike the Witten index $\Omega(\gamma, y, \zeta)$, the indices $\Omega_S(\gamma, y)$ are independent of the stability parameters $\zeta$. This property also holds for the attractor indices $\Omega_\star(\gamma, y)$ introduced in [18], but contrary to attractor indices[1], single-centered indices do not

---

[1]The attractor indices $\Omega_\star(\gamma, y)$ are instances of the Witten index in a particular chamber known as the attractor

yet have a first principle definition, nor a representation theoretic underpinning.[2] The general prescription for recovering $\Omega(\gamma, y, \zeta)$ in terms of the single-centered indices $\Omega_S(\gamma, y)$ has been called the Coulomb branch formula (see [14] for a concise review) and was tested in many examples [12, 17, 23], but it has remained conjectural in general. A notable exception is the case of quivers without oriented cycles, where all $\Omega_S$'s vanish except those associated to the basis vectors $\gamma_i$ [13].

Our aim in this work is to revisit this heuristic prescription, and derive the sum over collinear configurations and its modifications from the rigorous localization analysis of [4] in the full QQM. While the the Jeffrey-Kirwan residue prescription in this reference was obtained by applying Stokes' theorem to an integral over the complexified Cartan torus *and* auxiliary fields, we shall instead evaluate this integral by steepest descent,[3]. In the absence of oriented cycles, we show that this yields a sum over collinear fixed points, exactly as specified in [10]. In the presence of oriented cycles, we shall show that there are additional saddle points, which are solutions to a deformed version of (1.2),

$$\forall i, \quad \sum_{j \neq i} \frac{\kappa_{ij}}{|z_i - z_j - \frac{\pi \mathrm{Im} z}{\beta} R_{ij}|} = 2\zeta_i, \tag{1.3}$$

where $R_{ij}$ is the R-charge of the $\kappa_{ij}$ chiral fields charged under $U(1)_i \times U(1)_j$ (chosen such that oriented cycles have R-charge 2, and defined for all $i, j$ such that $R_{ji} = -R_{ij}$), $\beta$ is the inverse temperature and $z$ (not to be confused with positions of the centers) is related to the angular momentum fugacity by $y = e^{i\pi z}$. Among these solutions, some (dubbed as regular collinear solutions) smoothly merge on solutions to (1.2) as $\beta \to \infty$, while others (dubbed as collinear scaling solutions) have no counterpart in (1.2), but persist for any value the FI parameters. In particular, they are the only ones remaining in the 'deep scaling regime' $\zeta_i \to 0$ considered in [25, 26]. We show that these collinear scaling solution complement the regular ones exactly as specified by the 'minimal modification hypothesis' of [11], and in addition provide the missing single-centered (or pure-Higgs) contribution. We demonstrate this in the case of Abelian cyclic quivers, where the equations (1.3) for $\zeta_i = 0$ can be solved explicitly. In that case, we further observe that the so-called 'same sign' scaling solutions exactly reproduce the stacky invariants with trivial stability condition – a mathematically well-defined notion (see e.g. [27]), but whose physics significance had remained obscure. Unfortunately, we do not yet understand the mathematical significance of the remaining 'unequal sign' scaling solutions.

The remainder of this article is organized as follows. In §2, we give a brief review of $\mathcal{N} = 4$ quiver quantum mechanics, discuss the conditions for existence of scaling solutions, and recall the Coulomb Branch Formula in this context. In §3, we recall the localization computation of the refined Witten index, assemble some useful formulae for dealing with infinite products appearing in this computation, and evaluate the Witten index for quivers using steepest descent, both in the absence (§3.3) and presence (§3.4) of oriented cycles. The case of a 3-node quiver is discussed in §3.4.1. In §4, we consider Abelian cyclic quivers with arbitrary number of nodes, and obtain generating series for a variety of indices including single-centered indices, attractor indices, trivial stability indices and scaling indices. Some further computational details are relegated to appendices.

---

or self-stability chamber. The Witten index in any chamber can be recovered from those by using attractor flow tree formulae [18–22].

[2]It is expected that single-centered indices count harmonic forms in the middle cohomology of the Higgs branch, but their precise characterization has been elusive.

[3]This is similar in spirit to the approach developped in [24] in the simplest case of the Kronecker quiver with rank $(1, N)$, but the details are different, and our method also applies to quivers with oriented loops.

# 2 Review of the Coulomb branch formula for quivers

In this section, we briefly review the quiver quantum mechanics (QQM) describing interactions of half-BPS dyons in supersymmetric field theories or string vacua with $\mathcal{N} = 2$ supersymmetry in 3+1 dimensions, and the Coulomb branch formula prescription for computing its index.

## 2.1 $\mathcal{N} = 4$ quiver quantum mechanics

QQM is a 0+1-dimensional gauge theory with $\mathcal{N} = 4$ supercharges [2], gauge group $G = \prod_{a=1}^{K} U(N_a)$ and $\kappa_{ab}$ chiral multiplets in the bifundamental representation of $U(N_a) \times U(N_b)$. Here $N_a$ are the number of constituents with distinct electromagnetic charges $\gamma_a$, and $\kappa_{ab}$ is the integer-valued Dirac-Schwinger-Zwanziger pairing.

It is convenient to view QQM as a special supersymmetric gauge theory with $\mathcal{N} = 2$ supercharges, where the couplings are tuned such as to enhance the supersymmetry to $\mathcal{N} = 4$. Recall that a $\mathcal{N} = 4$ vector multiplet decompose into $\mathcal{N} = 2$ vector multiplets and $\mathcal{N} = 2$ chiral multiplets:

$$(v_t, x_3, \lambda_-, D) \oplus (\sigma = x_1 + \mathrm{i}x_2, \mathrm{i}\bar{\lambda}_+). \tag{2.1}$$

We shall attach an index $ss'$ to the components of the multiplet in the $U(N_a)$ factor, with $s, s' = 1 \ldots N_a$. Under the Cartan torus $U(1)^r \subset G$ with $r = \sum_{a=1}^{K} N_a$, the off-diagonal components of the complex scalar field $\sigma_a^{s,s'}$ carry charge vector $V_a^{s,s'} := e_{a,s} - e_{a,s'}$, where the vector $e_{a,s}$ has component $+1$ along the direction $\alpha = (a, s)$ inside $U(1)^r$, and $0$ along the other directions. The $N = 4$ chiral multiplets decompose into $\mathcal{N} = 2$ chiral multiplets and $\mathcal{N} = 2$ Fermi multiplets in the same representation,

$$(\phi, \psi^+) \oplus (\psi^-, F). \tag{2.2}$$

Since they transform in the bifundamental representation of $U(N_a) \times U(N_b)$, we shall attach an index $abss'$ to the components of the multiplet in the $U(N_a)$ factor, with $s = 1 \ldots N_a, s' = 1 \ldots N_b$. Under the Cartan torus $U(1)^r$, the complex scalar fields $\phi_{ab}^{i,ss'}$ carry charge vector $\tilde{V}_{ab}^{ss'} = e_{a,s} - e_{b,s'}$ Note that the diagonal $U(1) \subset G$ acts trivially, so the rank of the effective gauge group is $\ell = r - 1$, and we can omit one component (say the last) in the charge vectors $V$ and $\tilde{V}$, such that the $\alpha$ index runs only from 1 to $\ell = r - 1$.

The Lagrangian of the $\mathcal{N} = 2$ gauge theory depends on the coefficients for the standard kinetic terms of the vector and chiral multiplets, on the Fayet-Iliopoulos parameters $\zeta_a$ subject to the condition $\sum_a N_a \zeta_a = 0$, as well as on a choice of superpotentials $E$ and $J$, which are vector-valued holomorphic functions of the complex scalar fields $\sigma, \phi$ in the $\mathcal{N} = 2$ chiral multiplets, subject to the condition that $Tr(J \cdot E) = 0$. In order to enforce $\mathcal{N} = 4$ supersymmetry, we choose equal coefficients for the kinetic terms of the components (2.1) and (2.2) inside the $\mathcal{N} = 4$ vector and chiral multiplets, and take

$$E(\sigma, \phi) = \sigma \phi, \quad J(\sigma, \phi) = -dW(\phi), \tag{2.3}$$

where $W$ is a gauge invariant, holomorphic function of the scalars in the $\mathcal{N} = 4$ chiral multiplets only. We shall assume that $W$ is a linear combination of traces of products of these scalars along oriented cycles of the loop, with generic coefficients such that the F-term equations $\partial_\phi W = 0$ are independent away from the locus where all $\phi$'s vanish.

With this matter content, the QQM has $SU(2)_+ \times SU(2)_-$ global R-symmetry, with Cartan torus $U(1)_+ \times U(1)_-$ acting with the following charge assignments

|       | $v_t$ | $x_3$ | $\lambda_-$ | $D$ | $\sigma$ | $\bar{\lambda}_+$ | $\phi$ | $\psi^+$ | $\psi^-$ | $F$ |
|-------|-------|-------|-------------|-----|----------|-------------------|--------|----------|----------|-----|
| $J_+$ | 0 | 0 | 1 | 0 | 1 | 0 | $\frac{R}{2}$ | $\frac{R}{2}-1$ | $\frac{R}{2}$ | $\frac{R}{2}-1$ |
| $J_-$ | 0 | 0 | 0 | 0 | $-1$ | $-1$ | $\frac{R}{2}$ | $\frac{R}{2}$ | $\frac{R}{2}-1$ | $\frac{R}{2}-1$ |

where $R$ stands for the charges $R_{ab}$ of the chiral fields $\phi_{ab}$ under the $U(1)_R$ generator $J_+ + J_-$ (for convenience, we define $R_{ba} = -R_{ab}$). These symmetries hold provided $W(\phi)$ transforms homogeneously with R-charge 2 (in particular, this ensures that the coupling $F\partial_\phi W$ in the Lagrangian is invariant). The $U(1)_- \subset SU(2)_-$ factor corresponds to spatial rotations of the system of interacting dyons; it is an R-symmetry for the full $\mathcal{N} = 4$ supersymmetry, but an ordinary global symmetry with respect to the $\mathcal{N} = 2$ subalgebra. In addition, by dimensional analysis the model is invariant under rescaling

$$t \to t/s, \quad \vec{x} \to s\vec{x}, \quad \lambda \to s^{3/2}\lambda, \quad D \to s^2 D, \quad \zeta \to \zeta/s, \quad \frac{1}{e^2} \to \frac{1}{e^2}/s^3, \quad \beta \to \beta/s. \quad (2.4)$$

If one instead keeps fixed the dimensionful parameters $\zeta$ and $1/e^2$ while scaling the fields inverse temperature as in (2.4) and taking the limit $s \to 0$, one expects the $\mathcal{N} = 4$ supersymmetry to be enhanced to the superconformal algebra $D(2,1;0)$ [25, 26].

## 2.2 Semi-classical vacua and BPS states

Semiclassically, the quiver quantum mechanics admits two branches of supersymmetric vacua [2]:

- On the Higgs branch, the gauge symmetry is broken to the $U(1)$ center by the vevs of the chiral multiplet scalars $\phi_{ab,A,ss'}$, which are subject to the D and F-term relations,

$$\sum_{\substack{b:\kappa_{ab}>0 \\ A=1...\kappa_{ab}}} \sum_{\substack{s'=1...b}} \phi^*_{ab,A,ss'} \phi_{ab,A,ts'} - \sum_{\substack{b:\kappa_{ab}<0 \\ A=1...|\kappa_{ab}|}} \sum_{\substack{s'=1...b}} \phi^*_{ba,A,s's} \phi_{ba,A,s't} = \zeta_a \delta_{st} \quad \forall\, a,s,t$$

$$\frac{W}{\partial \phi_{ab,A,ss'}} = 0 \quad \forall\, a,b,A,s,s', \quad (2.5)$$

where $1 \le a,b \le K$, $1 \le s,t \le N_a, 1 \le s' \le N_b, 1 \le A \le \kappa_{ab}$. As a result, the space of gauge inequivalent classical supersymmetric vacua coincides with the moduli space $\mathcal{M}_Q(\gamma,\zeta)$ of stable representations of the quiver $Q$, with stability conditions determined by the FI parameters $\zeta$. Quantum mechanically, BPS states on the Higgs branch are harmonic forms on $\mathcal{M}_Q(\gamma,\zeta)$, or equivalently Dolbeault cohomology classes.

- On the Coulomb branch the gauge symmetry is broken to the diagonal subgroup $U(1)^{\sum_{a=1}^K N_a}$ and all chiral multiplets as well as off-diagonal vector multiplets are massive. After integrating out these degrees of freedom, the diagonal part $\vec{x}_i$ of the scalars in the vector multiplets must be solutions to Denef's equations (1.1), with the index $i$ running over all $r = \sum_a N_a$ pairs $(a,s)$ with $s = 1\ldots N_a$, and the corresponding $\kappa_{ij}$ and $\zeta_i$ are equal to $\kappa_{ab}$ and $\zeta_a$, in such a way that $\sum_{i=1}^r \zeta_i = 0$. The space of solutions modulo common translations is a phase space $\mathcal{M}_n(\{\kappa_{ij}, c_i\})$ of dimension $2n-2$, equipped with a natural symplectic form [28], invariant under $SO(3)$ rotations in $\mathbb{R}^3$ generated by the angular momentum

$$\vec{J} = \frac{1}{2} \sum_{i<j} \kappa_{ij} \frac{\vec{x}_i - \vec{x}_j}{|\vec{x}_i - \vec{x}_j|}. \quad (2.6)$$

Quantum mechanically, BPS states are harmonic spinors for the natural Dirac operator on $\mathcal{M}_n(\{\kappa_{ij}, \zeta_i\})$, and fit into multiplets of $SO(3)$ [28, 29].

For quivers without oriented cycles, the Higgs branch and Coulomb branch give two equivalent descriptions of the same quantum mechanical system, and have isomorphic BPS spectra, with the action of $SO(3)$ rotations on the Higgs branch side via the Lefschetz action on the

cohomology of $\mathcal{M}_Q$. In contrast, for quivers with oriented cycles, the Coulomb branch description is incomplete, due to the fact that there exists loci on the phase space $\mathcal{M}_n(\{\kappa_{ij}, \zeta_i\})$ where the vectors $\vec{x}_i$ become arbitrarily close and the chiral fields become almost massless, such that it is no longer legitimate to integrate them out. These singular solutions are known as scaling solutions, and they exist under certain conditions on the arrow degeneracies $|\kappa_{ab}|$ which we review in the next subsection.

In order to count supersymmetric ground states, keeping track of the angular momentum of the corresponding BPS bound states in $D = 3 + 1$, it is convenient to consider the refined Witten index

$$\mathcal{I} = \mathrm{Tr}_{\mathcal{H}}(-1)^F e^{2\pi i z J_-} e^{-\beta H}, \tag{2.7}$$

where $H$ is the Hamiltonian of QQM, $F$ is the fermion number and the chemical potential $z$ is related to the usual fugacity by $y = e^{i\pi z}$. When $H$ is gapped (which holds for generic values of the FI parameters $\zeta = (\zeta_1, \ldots, \zeta_K)$ and coprime dimension vector $\gamma = (N_1, \ldots, N_K)$, the index (2.7) is independent of $\beta$, and computes the $\chi_{y^2}$-genus of the moduli space $\mathcal{M}_Q(\gamma, \zeta)$ of stable representations,

$$\mathcal{I} = \Omega(\gamma, y, \zeta) := \sum_{p=0}^{d} h_{p,q}(\mathcal{M})(-1)^{p+q-d} y^{2q-d}, \tag{2.8}$$

where $d$ is the complex dimension of $\mathcal{M} = \mathcal{M}_Q(\vec{N}, \vec{\zeta})$. Put differently, $\mathcal{I}$ gives a weighted count of BPS states in the Higgs branch description. It also counts BPS states on the Coulomb branch whenever the latter is well-defined, i.e. in the absence of scaling solutions.

When the dimension vector $\gamma$ is not primitive, the Witten index (2.7) instead computes the rational index [7].

$$\mathcal{I} = \bar{\Omega}(\gamma, y, \zeta) := \sum_{d|\gamma} \frac{y - 1/y}{d(y^d - y^{-d})} \Omega(\gamma/d, y^d, \zeta), \tag{2.9}$$

where $\Omega$ on the r.h.s. is defined as in (2.8), using $L^2$-cohomology when $\mathcal{M}$ is non-compact.

## 2.3 Scaling solutions

For scaling solutions such that all $\vec{x}_i$ become nearly concident, the FI parameters on the r.h.s. of (1.1) become irrelevant, and the equations reduce to the 'conformal Denef equations',

$$\forall i, \quad \sum_{j \neq i} \frac{\kappa_{ij}}{|\vec{x}_i - \vec{x}_j|} = 0. \tag{2.10}$$

If they exist, they occur in one-parameter families where all distances are scaled by a factor of $\lambda > 0$. Since the angular momentum (2.6) on solutions to (1.1) evaluates to $\vec{J} = 2 \sum_i \zeta_i \vec{x}_i$, it follows that scaling solutions carry vanishing angular momentum at the classical level. In particular, collinear solutions to (2.10)[4] exist only for nongeneric values of the $\kappa'_{ij}s$ such that $\sum_{i<j} \kappa_{\sigma(i),\sigma(j)}$ vanishes for some permutation $\sigma$.

For $K = 3$ centers, it is clear that solutions to (2.10) exist if and only if $\kappa_{12}, \kappa_{23}, \kappa_{31}$ have the same sign (positive, say) and satisfy the triangular inequality

$$\kappa_{12} + \kappa_{23} \geq \kappa_{31} \tag{2.11}$$

and cyclic permutations thereof. These inequalities ensure that $r_{ij} = \lambda \kappa_{ij}$ correspond to the distances between an actual configuration of 3 points in $\mathbb{R}^3$. We conjecture that a necessary

---

[4]We reserve the phrase 'collinear scaling solutions' for solutions of the deformed equations (1.3).

condition for any number of centers $K$ such that the nodes $1, 2, \ldots K$ form an oriented cycle on the quiver is that[5]

$$\sum_{i<j} \kappa_{ij} \geq 0 \quad \text{and cyclic perm.}. \tag{2.12}$$

In the case of a cyclic quiver, with $\kappa_{ij} = 0$ unless $j = i+1$ (with $j = K+1$ identified with $j = 1$) this condition reduces to

$$0 < \kappa_{K,1} < \kappa_{12} + \kappa_{23} + \cdots + \kappa_{K-1,K} \tag{2.13}$$

and cyclic permutations thereof, which is again a trivial consequence of the fact that $r_{i,i+1} = \lambda \kappa_{i,i+1}$ correspond to distances between $K$ points in $\mathbb{R}^3$. For a cyclic quiver with one additional arrow, say $\kappa_{1,k} > 0$ with $k \neq 2$ and $k \neq K$, one may also demonstrate that (2.12) is a necessary condition (see Appendix A). We do not know how to show that (2.12) holds in general, but we observe that this is the most general condition which is linear in the $\kappa_{ij}$'s, and which reduces to the known conditions for a cyclic quiver with one additional arrow. For $K = 4$, using the results in ( [18, (4.15)], we can prove by a case-by-case analysis that (2.12) is a necessary condition for the non-vanishing of the difference $\Omega_\star(\gamma) - \Omega_S(\gamma)$, see below).

Quantum mechanically, we conjecture that the condition for existence of scaling bound states is strengthened to

$$\sum_{i<j} \kappa_{ij} \geq K-1 \quad \text{and cyclic perm.}, \tag{2.14}$$

generalizing the known condition for cyclic quivers [12]. This condition is consistent with the positivity of the expected dimension of the Higgs branch in a chamber where all chiral fields $\Phi_{ij}^\alpha$ with $i > j$ vanish.

## 2.4 The Coulomb branch formula

For quivers without oriented cycles, the Coulomb branch formula expresses the rational index $\overline{\Omega}(\gamma, y, \zeta)$ defined in (2.9) as a sum over all possible unordered decompositions of the dimension vector $\gamma$ into a sum of positive dimension vectors $\alpha_i$,

$$\overline{\Omega}(\gamma, y, \zeta) = \sum_{\gamma = \sum_{i=1}^n \alpha_i} \frac{g_C(\{\alpha_{ij}, c_i\}, y)}{|\text{Aut}\{\alpha_i\}|} \prod_{i=1}^n \bar{\Omega}_S(\alpha_i, y). \tag{2.15}$$

Here the rational indices $\overline{\Omega}_S(\alpha_i, y)$ are zero except when the dimension vector $\alpha_i$ has support on only one node of the quiver, in which case $\overline{\Omega}_S(\alpha_i, y) = (y - 1/y)/[N(y^N - y^{-N})]$ where $N$ is the value of $\alpha$ on that node. Said differently, the integer indices $\Omega_S(\alpha_i, y)$ defined as in (2.9) are equal to one if $\alpha_i$ is the unit dimension vector on one node of the quiver, or zero otherwise. The factor $|\text{Aut}\{\alpha_i\}|$ is the usual Boltzmann symmetry factor, i.e. the order of the subgroup of permutations of $n$ elements which preserve the ordered list $\{\alpha_i, i = 1 \ldots n\}$.

The coefficient $g_C(\{\alpha_i, c_i\}, y)$, known as the Coulomb index, is the equivariant index of the Dirac operator on the phase space $\mathcal{M}_n(\{\alpha_{ij}, c_i\})$, where $\alpha_{ij} = \langle \alpha_i, \alpha_j \rangle$ and $c_i = (\zeta, \alpha_i)$ are the FI parameters associated to the constituents. The index was computed by localization with respect to rotations around a fixed axis in [10, 11, 13]. The fixed points of the action of

---

[5]In case there are several oriented cycles passing through all the nodes, these conditions are to be imposed for each such cycle. In case there is no oriented cycle passing through all the nodes, one should perturb the matrix $\kappa_{ij}$ such that such a cycle is created.

$J_3$ on $\mathcal{M}_n(\{\alpha_{ij}, c_i\})$ are collinear black hole solutions, with coordinates $z_i$ satisfying the one-dimensional Denef equations (1.2) (with $\kappa_{ij}, \zeta_i$ replaced by $\alpha_{ij}, c_j$). Denoting by $S$ the set of such solutions, up to overall translations, one has

$$g_C(\{\gamma_i, c_i\}, y) = \frac{(-1)^{n-1+\sum_{i<j}\gamma_{ij}}}{(y-y^{-1})^{n-1}} \sum_{s \in S} \operatorname{sgn}(\det{}' \partial_i \partial_j \mathcal{W}) \, y^{\sum_{i<j} \kappa_{\sigma(i)\sigma(j)}}, \qquad (2.16)$$

where $\partial_i \partial_j \mathcal{W}$ denotes the Hessian of the function

$$\mathcal{W}(\{z_i\}) = -\frac{1}{2} \sum_{i<j} \operatorname{sgn}(z_j - z_i) \alpha_{ij} \log|z_i - z_j| - \sum_i c_i z_i. \qquad (2.17)$$

When the phase space $\mathcal{M}_n(\{\alpha_{ij}, c_i\})$ is compact, which is the case for quivers without oriented cycles, the sum over fixed points produces a symmetric Laurent polynomial in $y$, which is the character of the $SO(3)$ representation spanned by harmonic spinors.

While the formula (2.15) for quivers without oriented cycles is transparent and well established [13], its generalization to quivers with oriented cycles is conjectural and more involved [11, 13, 14]:

$$\overline{\Omega}(\gamma, y, \zeta) = \sum_{\gamma = \sum_{i=1}^n \alpha_i} \frac{g_C(\{\alpha_{ij}, c_i\}, y)}{|\operatorname{Aut}\{\alpha_i\}|} \prod_{i=1}^n \bar{\Omega}_T(\alpha_i, y), \qquad (2.18)$$

where $\bar{\Omega}_T(\alpha_i, y)$ is constructed in terms of $\Omega_T(\alpha_i, y)$ by a relation similar to (2.9). The factor $g_C(\{\alpha_{ij}, \zeta_i\}, y)$ is defined by (2.16) just as in the previous case, however it is in general not a symmetric Laurent polynomial in $y$, due to the fact that the phase space $\mathcal{M}_n(\{\alpha_{ij}, c_i\})$ is not compact. Indeed, it misses the scaling solutions, which carry zero angular momentum and should therefore contribute to the sum over fixed points.

As for the 'total' invariant $\Omega_T(\alpha, y)$, it is in turn determined in terms of the single-centered indices $\Omega_S(\beta_j, y)$ via

$$\Omega_T(\alpha, y) = \Omega_S(\alpha, y) + \sum_{\alpha = \sum_{i=1}^m m_i \beta_i} H(\{\beta_i, m_i\}, y) \prod_{i=1}^m \Omega_S(\beta_i, y^{m_i}), \qquad (2.19)$$

where the sums run over unordered decompositions of $\alpha$ into sums of vectors $m_i \beta_i$ with $m_i \geq 1$ and $\beta_i$ a linear combination of the $\alpha_a$'s with positive integer coefficients[6]. Unlike in the absence of oriented cycles, the single-centered indices $\Omega_S(\alpha, y)$ may be non-vanishing on any dimension vector $\alpha$ whose support spans oriented cycles – in addition to the basic dimension vectors associated to each node, for which $\Omega_S(\alpha, y) = 1$.

The functions $H(\{\beta_i, m_i\}, y)$ are supposed to incorporate the missing contributions of scaling fixed points to the equivariant index of $\mathcal{M}_n(\{\alpha_{ij}, c_i\})$. While it is not known yet how to compute them from first principles, an ad hoc prescription, called 'minimal modification hypothesis, was put forward in [13, 14]. This prescription amounts to replacing the coefficient $f(y)$ of $\prod_{i=1}^m \Omega_S(\beta_i, y^{m_i})$, which is in general a rational function, by its image under the projection operator [12, (2.9)],

$$M[f](y) = \oint_0 \frac{du}{2\pi i} \frac{(1/u - u) f(u)}{(1 - uy)(1 - u/y)}, \qquad (2.20)$$

which turns a rational function into a symmetric Laurent polynomial with the same polar terms in the Laurent expansion at $y = 0$ or $y = \infty$,

$$f(y) = \sum_{n \geq -N} f_n y^n \rightarrow M[f](y) = \sum_{-N \leq n < 0} f_n(y^n + y^{-n}) + f_0. \qquad (2.21)$$

---

[6] If one of the constituents $\beta_i$ is not primitive, all choices $(dm_i, \beta_i/d)$ are counted as distinct contributions.

Note that the dependence on FI parameters is entirely contained in the Coulomb indices (2.16).

We note that the relations (2.9), (2.18), (2.19) are consistent with assigning a charge $m\gamma$ to the indices $\Omega(\gamma, y^d)$, $\Omega_S(\gamma, y^d)$ and their rational counterparts. Therefore, the index can be written in either of the two forms

$$\overline{\Omega}(\gamma, y, \zeta) = \sum_{\gamma=\sum_{i=1}^n m_i \alpha_i} \frac{\overline{g}_C(\{\alpha_i, m_i, \zeta_i\}, y)}{|\text{Aut}\{\alpha_i, m_i\}|} \prod_{i=1}^n \bar{\Omega}_S(\alpha_i, y^{m_i}), \qquad (2.22)$$

$$\Omega(\gamma, y, \zeta) = \sum_{\gamma=\sum_{i=1}^n m_i \alpha_i} \frac{\widehat{g}_C(\{\alpha_i, m_i, \zeta_i\}, y)}{|\text{Aut}\{\alpha_i, m_i\}|} \prod_{i=1}^n \Omega_S(\alpha_i, y^{m_i}), \qquad (2.23)$$

where the sum runs over unordered decompositions $\gamma = \sum_{i=1}^n m_i \gamma_i$ with $m_i \geq 1$, and $\text{Aut}\{\gamma_i, m_i\}$ denotes the subgroup of $S_n$ which preserves the pairs $(\gamma_i, m_i)$. We recall that the single centered indices $\Omega_S(\gamma, y)$ can be related to the attractor indices $\overline{\Omega}_\star(\gamma, y)$ by evaluating (2.22) at the attractor point $\zeta_i^\star = -\sum_j \kappa_{ij} N_j$ [18].

### 2.4.1 Abelian quivers

Assuming that $\gamma$ is a linear combination of the basis vectors $\gamma_a$ with coefficients at most 1, the Coulomb branch formula (2.18) simplifies to

$$\Omega(\gamma, y, \zeta) = \sum_{\gamma=\sum_{i=1}^n \alpha_i} \widehat{g}_C(\{\alpha_i, \zeta_i\}, y) \prod_{i=1}^n \Omega_S(\alpha_i, y), \qquad (2.24)$$

where

$$\widehat{g}_C(\{\gamma_i, \zeta_i\}, y) = \sum_{\substack{\sum_{i=1}^n \gamma_i = \sum_{i=1}^m \alpha_i \\ \alpha_i = \sum_{j=1}^{m_i} \beta_{i,j}}} g_C(\{\alpha_i, \zeta_i\}, y) \prod_{i=1}^m H(\{\beta_{i,1}, \ldots, \beta_{i,m_i}\}, y), \qquad (2.25)$$

with the understanding that $H(\{\beta_1\}, y) = 1$ and $H(\{\beta_1, \beta_2\}, y) = 0$. The term $H(\alpha_1, \ldots, \alpha_n, y)$ with the largest number of arguments is fixed by the minimal modification hypothesis, and is independent of the moduli.

## 3 Coulomb branch localisation

After reviewing the result of the localization computation in [4], we evaluate the integral using steepest descent, and recover the Coulomb branch prescription for Abelian quivers, both in the absence (§3.3) and presence (§3.4) of oriented cycles. The case of a 3-node quiver is discussed in §3.4.1.

### 3.1 Witten index from localisation

As explained in [5–7] and especially in [4], the Witten index (2.7) can be computed by localization, similar to the case of two-dimensional supersymmetric gauge theories analyzed in [9,30,31]. This procedure relies on the fact that the kinetic terms for the $\mathcal{N} = 2$ multiplets are $Q$-exact, and therefore the functional integral is independent of the values of the kinetic couplings $e$ and $g$. In the limit $e \to 0$, the integral localizes on configurations where the $x_3$'s are restricted to a common Cartan subalgebra, $x_{3,a}^{ss'} = \delta^{ss'} x_{3,a}$, and moreover $x_{3,a}$ is covariantly constant,

$$\nabla_t x_3 = 0, \qquad (3.1)$$

where $\nabla_t = \partial_t + i v_t$ is the gauge covariant derivative. In this limit, the one-loop approximation of the functional integral around configurations with constant $u := \frac{\beta}{2\pi}(v_t - i x_3)$ becomes exact, and (2.7) reduces to a finite dimensional integral

$$\mathcal{I} = \left(\frac{\beta^2}{4\pi^2}\right)^\ell \int \frac{\mathrm{d}^{2\ell} u \, \mathrm{d}^\ell D}{(2\pi)^\ell |W|} \, g(u, D) \det h(u, D) \, e^{-\beta S(D, \zeta)}, \tag{3.2}$$

where

$$S(D, \zeta) = \frac{1}{2e^2} \sum_{\substack{a=1\ldots K \\ s=1\ldots N_a}} D_{a,s}^2 - i \sum_{\substack{a=1\ldots K \\ s=1\ldots N_a}} \zeta_a D_{a,s} \tag{3.3}$$

and $|W| = \prod_a N_a!$ stands for the order of the Weyl group. In (3.2), the integral runs over the complex variables $u_\alpha = u_{a,s}$ and real variables[7] $D_\alpha = D_{a,s}$ with $\alpha = 1 \ldots \ell$. while the last entry is fixed by the conditions $\sum_{\alpha=1}^r u_\alpha = \sum_{\alpha=1}^r D_\alpha = 0$. The factor $g(u, D)$ is a product of one-loop determinants,

$$g(u, D) = \sin \pi z \prod_{a=1}^K g_{\text{vector}}^{(N_a)}(u_a, D_a) \prod_{\substack{a,b=1\ldots K \\ \kappa_{ab} > 0}} \left[ g_{\text{chiral}}^{(N_a, N_b)}(u_a, u_b, D_a, D_b) \right]^{\kappa_{ab}}, \tag{3.4}$$

where $g_{\text{vector}}$ is the contribution of a $\mathcal{N} = 4$ vector multiplet transforming in the adjoint representation of $U(N)$,

$$g_{\text{vector}}^{(N)}(u, D) = \frac{1}{(\sin \pi z)^N} \prod_{\substack{s,s'=1\ldots N \\ s \neq s'}} \prod_{m \in \mathbb{Z}} \frac{(m + u_s - u_{s'})(m + \bar{u}_{s'} - \bar{u}_s + \bar{z})}{|m + u_{s'} - u_s + z|^2 - \frac{i\beta^2}{4\pi^2}(D_s - D_{s'})}, \tag{3.5}$$

while $g_{\text{chiral}}^{(N,N')}$ is the one-loop determinant for a $\mathcal{N} = 4$ chiral multiplet transforming in the bifundamental representation of $U(N) \times U(N')$,

$$g_{\text{chiral}}^{(N,N')}(u, u', D, D') = \prod_{s=1}^N \prod_{s'=1}^{N'} \prod_{m \in \mathbb{Z}} \frac{\left(m + \bar{u}'_{s'} - \bar{u}_s - \frac{1}{2}R\bar{z}\right)\left(m + u_s - u'_{s'} + (\frac{1}{2}R - 1)z\right)}{|m + u'_{s'} - u_s - \frac{1}{2}Rz|^2 - \frac{i\beta^2}{4\pi^2}(D_s - D'_{s'})}. \tag{3.6}$$

The factor $\sin \pi z$ in (3.4) comes as a result of removing the diagonal $U(1)$ factor in $G$. Finally, $h$ is a $\ell \times \ell$ symmetric matrix coming from saturating the gaugino fermionic zero-modes,

$$h_{\alpha\beta}(u, D) = \sum_{\substack{a=1\ldots K \\ s,s'=1\ldots N_a \\ s \neq s'}} \sum_{m \in \mathbb{Z}} \frac{V_{a,\alpha}^{s,s'} V_{a,\beta}^{s,s'}}{\left(m + \bar{u}_{a,s'} - \bar{u}_{a,s} + \bar{z}\right)\left[|m + u_{a,s'} - u_{a,s} + z|^2 - \frac{i\beta^2}{4\pi^2}(D_{a,s} - D_{a,s'})\right]}$$

$$+ \sum_{\substack{a,b=1\ldots K \\ \kappa_{ab} > 0 \\ s=1\ldots N_a \\ s'=1\ldots N_b}} \sum_{m \in \mathbb{Z}} \frac{\tilde{V}_{a,b,\alpha}^{s,s'} \tilde{V}_{a,b,\beta}^{s,s'}}{\left(m + \bar{u}_{b,s'} - \bar{u}_{a,s} - \frac{1}{2}R_{ab}\bar{z}\right)\left[|m + u_{b,s'} - u_{b,s} - \frac{1}{2}R_{ab}z|^2 - \frac{i\beta^2}{4\pi^2}(D_{a,s} - D_{b,s'})\right]}, \tag{3.7}$$

where $\tilde{V}_{a,\alpha}^{s,s'}$ and $\tilde{V}_{a,b,\alpha}^{s,s'}$ are the components of the charge vectors for $\mathcal{N} = 2$ chiral multiplets defined in the previous subsection. An important property of (3.4) is

$$\partial_{\bar{u}_\alpha} g(u, D) = -\frac{i\beta^2}{4\pi^2} h_{\alpha\beta}(u, D) D^\beta \, g(u, D). \tag{3.8}$$

---

[7]The variable $D$ is related to the auxiliary field $\mathcal{D}$ by a factor $\beta^2/(4\pi^2)$, which accounts for the prefactor.

As explained in [4, 9], using the identity (3.8) the integral over $u, \bar{u}$ can be cast into a contour integral in the $u$-plane, and the integral over $D$ evaluated by computing the residue at $D = 0$. The remaining contour integral over $u$ leads to a sum over residues, with a precise prescription for determining which of them contribute for given value of the FI parameters $\zeta$, known as the Jeffrey-Kirwan residue. Instead, in order to make contact with the heuristic localization on the Coulomb branch, we shall evaluate the integral over $u, \bar{u}, D$ by directly saddle point methods.

## 3.2 Evaluating infinite products and sums

Before proceeding, we shall evaluate the infinite products in (3.6) using trigonometric functions. At $D = 0$, the infinite product in (3.6) can be computed using the identity

$$\frac{\sin \pi x}{\pi x} = \prod_{m \neq 0} \left(1 - \frac{x}{m}\right), \tag{3.9}$$

leading to the well-known expression [4, 9],

$$g_{\text{chiral}}^{(N.N')}(u, u', 0, 0) = \prod_{s=1}^{N} \prod_{s'=1}^{N'} \frac{\sin \pi \left(u_s - u'_{s'} + (\frac{1}{2}R - 1)z\right)}{\sin \pi \left(u'_{s'} - u_s - \frac{1}{2}Rz\right)}. \tag{3.10}$$

Similarly, the product (3.5) reduces to

$$g_{\text{vector}}^{(N)}(u, 0) = \frac{1}{(\sin \pi z)^N} \prod_{\substack{s, s'=1 \dots N \\ s \neq s'}} \frac{\sin[\pi(u_{s'} - u_s)]}{\sin[\pi(u_s - u_{s'} - z)]}. \tag{3.11}$$

For $D \neq 0$, the infinite product can be computed similarly using the identity

$$\frac{\cos 2\pi x - \cos 2\pi a}{1 - \cos 2\pi a} = \prod_{m \in \mathbb{Z}} \left(1 - \frac{x^2}{(m+a)^2}\right), \tag{3.12}$$

which holds since both sides vanish whenever $x = \pm a + m$ with $m \in \mathbb{Z}$. More generally,

$$\prod_{m \in \mathbb{Z}} \left(1 - \frac{x^2}{(m+a)^2 + b^2}\right) = \prod_{m \in \mathbb{Z}} \frac{1 - \frac{x^2 - b^2}{(m+a)^2}}{1 + \frac{b^2}{(m+a)^2}} = \frac{\cosh 2\pi\sqrt{b^2 - x^2} - \cos 2\pi a}{\cosh 2\pi b - \cos 2\pi a}, \tag{3.13}$$

which reduces to the previous formula when $b \to 0$. Applying this identity to the ratio between the values at $D \neq 0$ and $D = 0$, we get we get

$$\frac{g_{\text{chiral}}^{(N,N')}(u, u', D, D')}{g_{\text{chiral}}(u, u', 0, 0)} = \prod_{s=1}^{N} \prod_{s'=1}^{N'} \prod_{m \in \mathbb{Z}} \frac{|m + u_s - u'_{s'} + \frac{1}{2}Rz|^2}{|m + u_s - u'_{s'} + \frac{1}{2}Rz|^2 - \frac{i\beta^2}{4\pi^2}(D_s - D'_{s'})}$$

$$= \prod_{s=1}^{N} \prod_{s'=1}^{N'} \frac{\cosh(2\pi \, \text{Im} U_{ss'}) - \cos(2\pi \, \text{Re} U_{ss'})}{\cosh(2\pi \sqrt{\text{Im}^2(U_{ss'}) - \frac{i\beta^2}{4\pi^2} D_{ss'}}) - \cos(2\pi \text{Re} U_{ss'})}$$

$$= \prod_{s=1}^{N} \prod_{s'=1}^{N'} \frac{\cosh \beta \Sigma_{ss'} - \cos \beta V_{ss'}}{\cosh(\beta \sqrt{\Sigma_{ss'}^2 - i D_{ss'}}) - \cos \beta V_{ss'}},$$

where

$$U_{ss'} = u_s - u'_{s'} + \frac{R}{2}z = \frac{\beta}{2\pi}(V_{ss'} - i\Sigma_{ss'}), \quad D_{ss'} = D_s - D'_{s'}. \tag{3.14}$$

The infinite sums in (3.7) can similarly be evaluated in terms of trigonometric functions by using the identity

$$\sum_{m\in\mathbb{Z}} \frac{1}{(m+a+ib)(m+a-ib)^2} = -\frac{i\pi^2}{2b[\sin\pi(a-ib)]^2} + \frac{\pi}{4b^2}\left[\cot\pi(a-ib)-\cot\pi(a+ib)\right]. \tag{3.15}$$

### 3.3 Abelian quivers without oriented cycles

In this section, we establish the Coulomb branch formula for quivers without oriented cycles. We note that the Coulomb branch formula for such quivers has been established previously in [13,32], by showing its equivalence with Reineke's formula. Our aim however is to explain physically the origin of the sum over collinear configurations. For simplicity, we start with Abelian quivers, before generalizing the argument to the non-Abelian case.

For ranks $N=N'=1$, (3.5) and (3.6) reduce to

$$
\begin{aligned}
g_{\text{vector}}(u,D) &= \frac{1}{\sin\pi z}, \\
g_{\text{chiral}}(u-u',D-D') &= \prod_{m\in\mathbb{Z}} \frac{\left(m+\bar{u}-\bar{u}'+\frac{1}{2}R\bar{z}\right)\left(m+u-u'+(\frac{1}{2}R-1)z\right)}{|m+u-u'+\frac{1}{2}Rz|^2-\frac{i\beta^2}{4\pi^2}(D-D')}.
\end{aligned} \tag{3.16}
$$

The infinite product in (3.16) can be evaluated using the identity (3.14). In the regime where $\beta|\Sigma|\gg 1$ and $D$ is of order $|\Sigma|$ or less, the hyperbolic cosine in (3.14) dominates over the trigonometric cosine, so that we can approximate

$$g_{\text{chiral}}(u_a-u_b,D_a-D_b) \sim e^{-\beta\sqrt{\Sigma_{ab}^2-iD_{ab}}+\beta|\Sigma_{ab}|} g_{\text{chiral}}(u_a-u_b,0), \tag{3.17}$$

where following (3.14), we have set

$$\Sigma_{ab}=\Sigma_a-\Sigma_b-\frac{\pi\text{Im}z}{\beta}R_{ab}, \quad \Sigma_a=-\frac{2\pi}{\beta}\text{Im}u, \quad D_{ab}=D_a-D_b. \tag{3.18}$$

The integral (3.2) therefore reduces to

$$\mathcal{I}=\left(\frac{\beta^2}{4\pi^2}\right)^\ell \int \frac{d^{2\ell}u\, d^\ell D}{(2\pi\sin\pi z)^\ell}\left(\prod_{\kappa_{ab}>0}\left[g_{\text{chiral}}(u_a-u_b,0)\right]^{\kappa_{ab}}\right)\det h(u,D)\,e^{-\beta S(D,\Sigma)}, \tag{3.19}$$

where

$$S(D,\Sigma)=\frac{1}{2e^2}\sum_a D_a^2+i\sum_a \zeta_a D_a-\sum_{\kappa_{ab}>0}\kappa_{ab}\left(|\Sigma_{ab}|-\sqrt{\Sigma_{ab}^2-iD_{ab}}\right). \tag{3.20}$$

The action (3.20) is recognized as the one-loop effective action obtained in [2, (4.2)] after integrating out massive chiral multiplets, assuming that the transverse scalars $\sigma^a=x_1^a+ix_2^a$ in the vector multiplets have vanishing expectation value.

We shall now carry out the integral over $D$ using the saddle point method, which is valid in the limit $\beta\to\infty$. We make the self-consistent assumption that $|\Sigma_{ab}|^2\gg|D_{ab}|$ at the saddle point, so that the square root can be expanded:

$$S(D,\Sigma)\sim\frac{1}{2e^2}\sum_a D_a^2+i\sum_a \zeta_a D_a-\frac{i}{2}\sum_{\kappa_{ab}>0}\kappa_{ab}\frac{D_{ab}}{|\Sigma_{ab}|}. \tag{3.21}$$

Moreover, the prefactor $\det h(u, D)$ is polynomially bounded in this regime. Indeed, using (3.15) we find (identifying the indices $\alpha\beta$ and $ab$)

$$h_{ab} = -\frac{2i\pi^3}{\beta^2}\left(\delta_{ab}\sum_{c\neq a}\frac{\kappa_{ac}}{\Sigma_{ac}^2}\,\mathrm{sgn}\Sigma_{ac} - \frac{\kappa_{ab}}{\Sigma_{ab}^2}\,\mathrm{sgn}\Sigma_{ab}\right). \tag{3.22}$$

As $\beta \to \infty$, the integral over $D_a$ is therefore dominated by a saddle point at

$$D_a^\star = -i e^2\left(\zeta_a - \sum_{b\neq a}\frac{\kappa_{ab}}{2|\Sigma_{ab}|}\right), \tag{3.23}$$

where $\Sigma_{ab}$ was defined in (3.18). Integrating out $D$ in this manner, and neglecting higher loop corrections which are suppressed as $\beta \to \infty$, the integral (3.19) reduces to an integral over $u$,

$$\begin{aligned}\mathcal{I} &= \left(\frac{\beta^2}{4\pi^2}\right)^\ell\left(\frac{2\pi e^2}{\beta}\right)^{\ell/2}\int\frac{\mathrm{d}^{2\ell}u}{(2\pi i\sin\pi z)^\ell}\left(\prod_{\kappa_{ab}>0}[g_{\mathrm{chiral}}(u_a-u_b,0)]^{\kappa_{ab}}\right)\\ &\quad\times\;\det[ih(u,D^\star)]\,e^{\frac{\beta}{2e^2}\sum_a(D^\star)^2}.\end{aligned} \tag{3.24}$$

Let us now perform the integral over $\Sigma_a = -\frac{2\pi}{\beta}\mathrm{Im}u_a$. Assuming that $g_{\mathrm{chiral}}(u_a-u_b)$ varies slowly as a function of $\Sigma_a$, the integral is again dominated for large $\beta$ by saddle points where $D^\star(\Sigma) = 0$. For quivers without oriented cycles, the solutions to these equations are independent of $\beta$ in the limit $\beta \to \infty$, with $\kappa_{ab}/|\Sigma_{ab}|$ and $|\zeta_a - \zeta_b|$ of the same order. We may approximate $\Sigma_{ab} = \Sigma_a - \Sigma_b$, such that (3.23) becomes

$$D_a^\star = -i e^2\left(\zeta_a - \sum_{b\neq a}\frac{\kappa_{ab}}{2|\Sigma_a - \Sigma_b|}\right). \tag{3.25}$$

The conditions $\beta|\Sigma_{ab}| \gg 1$ and $|\Sigma_{ab}|^2 \gg |D_{ab}|$ require that both $\beta$ and $1/e^{2/3}$ are much greater than $|\zeta_a - \zeta_b|$, consistently with the scaling symmetry (2.4). In contrast, for quivers with oriented cycles, there exists another branch of solutions where $|\Sigma_{ab}|$ scales like $\mathrm{Im}z/\beta$ as $\beta \to \infty$, and $\Sigma_{ab}$ can no longer be identified with $\Sigma_a - \Sigma_b$. We postpone this issue to §3.4.

Assuming that the approximate version (3.25) of (3.23) is valid, the saddle points for the integral over $\Sigma_a$ must then satisfy

$$\forall a,\quad \sum_{b\neq a}\frac{\kappa_{ab}}{|\Sigma_a - \Sigma_b|} = 2\zeta_a. \tag{3.26}$$

This is recognized as the one-dimensional reduction of Denef's equations (1.1), which constrain the relative distances of multi-centered BPS black holes in 3+1 dimensions. Here, the centers are constrained to the $z$-axis where $x_1^a = x_2^b = 0$, and the solutions to (3.26) are discrete (except for overal translations $\Sigma^a \to \Sigma^a + \delta$). We denote by $S$ the set of such collinear solutions, indexed by the allowed orderings $s \in \subset S_{\ell+1}$ of the centers along the $z$-axis [10].

Since the solutions to (3.26) satisfy $\beta|\Sigma_{ab}| \gg \mathrm{Im}z$, the bracketed factor in (3.24) becomes independent of $u$ in the limit $\beta \to \infty$, justifying *a posteriori* our assumption that $g_{\mathrm{chiral}}$ varies slowly:

$$\prod_{\kappa_{ab}>0}\left[\frac{\sin(\pi(U_a-U_b-z))}{\sin(\pi(U_b-U_a))}\right]^{\kappa_{ab}} \sim (-1)^{\sum_{a<b}\kappa_{ab}}\,e^{i\pi z\sum_{a<b}\kappa_{ab}\,\mathrm{sgn}\Sigma_{ab}}. \tag{3.27}$$

This is recognized as the angular momentum factor $y^{2J_3(s)}$ weighting the collinear configurations in [10], where

$$J_3(s) = \frac{1}{2} \sum_{a<b} \kappa_{ab} \operatorname{sgn} \Sigma_{ab} = \frac{1}{2} \sum_{\kappa_{ab}>0} \kappa_{ab} \operatorname{sgn} \Sigma_{ab} \tag{3.28}$$

is the classical angular momentum carried by the configuration. The integral over $\operatorname{Re}(u) \in [0,1]$ is then trivial.

In order to carry out the integral around the saddle point at $D^\star(\Sigma) = 0$, we observe that the matrix $h_{\alpha\beta}$ can be reexpressed using (3.22) as

$$\mathrm{i}h_{ab}(u^\star, 0) \sim \frac{4\pi^3}{e^2 \beta^2} \partial_{\Sigma_a} \mathrm{i}D_b^\star = -\frac{4\pi^3}{\beta^2} \partial_{\Sigma_a} \partial_{\Sigma_b} \mathcal{W}, \tag{3.29}$$

where $\mathcal{W}$ is the 'superpotential' introduced in [10]

$$\mathcal{W} = -\frac{1}{2} \sum_{a<b} \operatorname{sgn}(\Sigma_b - \Sigma_a) \kappa_{ab} \log|\Sigma_a - \Sigma_b| - \sum_a \zeta_a \Sigma_a. \tag{3.30}$$

Thus, the factor $\det(\mathrm{i}h)$ on the locus $D^\star = 0$ is proportional to the Jacobian of the change of variables $\Sigma \to \mathrm{i}D^\star(\Sigma)$. Since the integral is Gaussian in terms of the variables $\mathrm{i}D^\star$, we obtain

$$
\begin{aligned}
\mathcal{I} &= \frac{(-1)^{\sum_{a<b} \kappa_{ab}}}{(2\mathrm{i}\pi \sin \pi z)^\ell} \left(\frac{\beta}{2\pi}\right)^{3\ell} \left(\frac{2\pi e^2}{\beta}\right)^{\ell/2} \left(\frac{4\pi^3}{e^2 \beta^2}\right)^\ell \\
&\quad \times \int \mathrm{d}^\ell \Sigma \, e^{\pi \mathrm{i}z \sum_{\kappa_{ab}>0} \kappa_{ab} \operatorname{sgn} \Sigma_{ab}} \det\left(\partial_{\Sigma_a} \mathrm{i}D_b^\star\right) e^{-\frac{\beta}{2e^2} \sum_a (\mathrm{i}D^\star)^2} \\
&= \frac{(-1)^{\sum_{a<b} \kappa_{ab}}}{(2\mathrm{i} \sin \pi z)^\ell} \left(\frac{\beta}{2\pi}\right)^{3\ell} \left(\frac{2\pi e^2}{\beta}\right)^\ell \left(\frac{4\pi^2}{e^2 \beta^2}\right)^\ell \sum_{s \in S} e^{2\pi \mathrm{i}z J_3(s)} \operatorname{sgn}(\det \partial_{\Sigma_a} \mathrm{i}D_b^\star) \\
&= \frac{(-1)^{\ell + \sum_{a<b} \kappa_{ab}}}{(y - 1/y)^\ell} \sum_{s \in S} y^{2J_3(s)} \operatorname{sgn}(\det \partial_a \partial_b \mathcal{W}),
\end{aligned}
\tag{3.31}
$$

where $s$ runs over collinear solutions to Denef's equations (3.26). The last line reproduces the prescription of [10] for computing the 'Coulomb index' $g_C(\{\gamma_a, \zeta_a\}, y)$ for $K$ distinguishable black holes carrying charges $\gamma_a$ such that $\kappa_{ab} = \langle \gamma_a, \gamma_b \rangle$. Since each center carries unit degeneracy $\Omega_S(\gamma_a) = 1$, this proves the simplest example of the Coulomb branch formula (2.15) for an Abelian quiver without oriented loops. In §3.4, we shall extend this analysis to the case of Abelian quivers with oriented cycles.

## 3.4 Abelian quivers with oriented cycles

As explained in §2.3, in the presence of oriented cycles and subject (conjecturally) to the inequalities (2.12), there exists non-compact regions where some of the centers can come arbitrarily close. In these regions, the previous analysis must be revisited, since the contribution proportional to $R_{ab}$ in (3.18) can no longer be ignored and the integrand is no longer independent of $\operatorname{Re}(u)$. We shall see that the first effect leads to a deformation of the saddle point equations (3.26), while the integral over $\operatorname{Re}(u)$ can, nonetheless, be evaluated exactly in the case of Abelian cyclic quivers.

For simplicity, we restrict to Abelian quivers with oriented cycles, although we do not yet impose that the conditions (2.12) for existence of classical scaling solutions are obeyed. Under the assumption that $\beta|\Sigma| \gg 1$, the analysis of §3.3 carries through up until (3.24). However, on performing the integral over $\Sigma_a$ in the saddle point approximation, we have to rely on the

full expression for $D_a^\star$ in (3.23), rather than its simplified version (3.25), since the former has additional solutions. As a result, we must look for solutions to the 'deformed Denef equations'

$$\forall a, \quad \sum_{b \neq a} \frac{\kappa_{ab}}{|\Sigma_a - \Sigma_b - \frac{\pi}{\beta} \mathrm{Im} z R_{ab}|} = 2\zeta_a. \tag{3.32}$$

In general, these equations admit two branches of solutions, distinguished by their behavior as $\beta \to \infty$: 'regular collinear solutions' where $|\Sigma_a - \Sigma_b| \sim 1/|\zeta_a - \zeta_b| \gg |\mathrm{Im} z|/\beta$, which are described by the usual Denef equations (3.25), and 'scaling collinear solutions', are such that $|\Sigma_{ab}|$ scales like[8] $|R_{ab} \mathrm{Im} z|/\beta$ as $\beta \to \infty$; for the latter, the $\mathrm{Im} z$ dependent term in the denominator can no longer be neglected. In §3.4.1, we demonstrate the existence of these two branches in the case of a cyclic three-node quiver.

The first class of solutions can be treated exactly as in §3.3 and leads to the same result (3.31),

$$\mathcal{I}_{\mathrm{reg}} = \frac{(-1)^{\ell + \sum_{a<b} \kappa_{ab}}}{(y - 1/y)^\ell} \sum_{s \in S_{\mathrm{reg}}} y^{2J_3(s)} \mathrm{sgn}(\det \partial_a \partial_b \mathcal{W}), \tag{3.33}$$

where $s$ runs over collinear solutions to Denef's equations (3.26) and $\mathcal{W}$ was defined in (3.30).

For the second class of solutions, we can effectively set $\zeta_a = 0$ in (3.32), for $a$ in a subset of nodes, which amounts to taking the 'deep scaling regime limit' introduced in [26]. Due to the non-vanishing value of $R_{ab}$, the solutions still form a discrete set which we denote by $S_{\mathrm{sc}}$. The integral over $\Sigma_a$ can still be performed in the saddle point approximation, but contrary to the previous case, the integral over $\mathrm{Re}(u)$ is no longer trivial due to the factors $g_{\mathrm{chiral}}$ in (3.24). As a result, we get

$$\mathcal{I}_{\mathrm{sc}} = \sum_{s \in S_{\mathrm{sc}}} \int_{[0,1]^\ell} \frac{(-1)^\ell \mathrm{d}^\ell \mathrm{Re}(u)}{(y - 1/y)^\ell} \left( \prod_{\kappa_{ab} > 0} [g_{\mathrm{chiral}}(u_a(s) - u_b(s), 0)]^{\kappa_{ab}} \right) \mathrm{sgn}(\det \partial_a \partial_b \widetilde{\mathcal{W}}),$$

where $\widetilde{\mathcal{W}}$ is the deformed version of (3.30),

$$\widetilde{\mathcal{W}} = -\frac{1}{2} \sum_{a<b} \mathrm{sgn}(\Sigma_b - \Sigma_a - \frac{\pi}{\beta} \mathrm{Im} z R_{ba}) \kappa_{ab} \log|\Sigma_a - \Sigma_b - \frac{\pi}{\beta} \mathrm{Im} z R_{ab}| - \sum_a \zeta_a \Sigma_a. \tag{3.34}$$

The integral over $\mathrm{Re}(u)$ (for fixed value of $\mathrm{Im} u$ determined by the solution $s \in S_{\mathrm{sc}}$) can be computed by Cauchy's theorem, or (as we do for cyclic quivers in §4.5) by Taylor expanding the various factors of $g_{\mathrm{chiral}}$. Note that (3.33) and (3.34) can be combined by summing over all (regular and scaling) collinear solutions in (3.34).

We claim that the sum of the two branches of collinear solutions reproduces the result predicted by the Coulomb branch formula:

$$\mathcal{I}_{\mathrm{reg}} + \mathcal{I}_{\mathrm{sc}} = (g_C(\{\gamma_a, \zeta_a\}, y) + H(\{\gamma_a\}, y)) \prod_{a=1}^K \Omega_S(\gamma_a, y) + \Omega_S\left( \sum_{a=1}^K \gamma_a, y \right). \tag{3.35}$$

In the next subsection §3.4.1 we shall demonstrate this agreement in the case of a 3-node cyclic quiver, postponing the case of a general cyclic quiver with $K$ nodes to §4.5. In §3.4.2 we analyze the deep scaling contribution for general Abelian quivers with multiple oriented cycles We have also implemented a Mathematica code (included in the package `CoulombHiggs.m` [33] which numerically solves the deformed Denef equations (3.32) and then computes the integral (3.34) (with the sum extended to all collinear solutions) by residue calculus. We have checked this algorithm correctly reproduces the index computed by the Jeffrey-Kirwan residue formula

---

[8]We note that this requires $|\mathrm{Im} z|$ to be large, but smaller than $\beta/|\zeta_a - \zeta_b|$.

in a variety of 4-node and 5 node quivers with more than one cycle (see the Mathematica notebook available from [33]). Unfortunately, the numerical search of solutions to (3.32) becomes impractical for $K > 5$, and it would be desirable to develop a recursive algorithm similar to [13] for assessing the existence of such solutions.

### 3.4.1 Three-node cyclic quiver

For a 3-node cyclic quiver with $a = \kappa_{12}, b = \kappa_{23}, c = \kappa_{31}$ with $a, b, c > 0$ and $R_{12} = R_{23} = R_{31} = 2/3$, the equations (3.32) become

$$
\begin{aligned}
\frac{a}{|\Sigma_1 - \Sigma_2 - r|} - \frac{c}{|\Sigma_3 - \Sigma_1 - r|} &= 2\zeta_1, \\
\frac{b}{|\Sigma_2 - \Sigma_3 - r|} - \frac{a}{|\Sigma_1 - \Sigma_2 - r|} &= 2\zeta_2, \\
\frac{c}{|\Sigma_3 - \Sigma_1 - r|} - \frac{b}{|\Sigma_2 - \Sigma_3 - r|} &= 2\zeta_3,
\end{aligned}
\tag{3.36}
$$

where we have set $r = \frac{2\pi}{3\beta}\mathrm{Im}z$. Without loss of generality, we work in the chamber $\zeta_1 > 0, \zeta_3 < 0$ and assume $r < 0$. Let us define

$$
\sigma_1 = \mathrm{sgn}(\Sigma_1 - \Sigma_2 - r), \quad \sigma_2 = \mathrm{sgn}(\Sigma_2 - \Sigma_3 - r), \quad \sigma_3 = \mathrm{sgn}(\Sigma_3 - \Sigma_1 - r).
\tag{3.37}
$$

For $r$ strictly zero and fixed signs $\sigma_i$, it was shown in [12, §3.1] that a solution $s_0$ exists whenever the sign of $\sigma_3$ is opposite to the sign of $a\sigma_1 + b\sigma_2 + c\sigma_3$. For this solution, the distances $|\Sigma_a - \Sigma_b|$ are of order $1/|\zeta_a - \zeta_b|$. We claim that for $r \neq 0$, there exists two branches of solutions:

- the 'regular' branch $s_{\mathrm{reg}}(r)$, which exists under the same condition $\sigma_3(a\sigma_1 + b\sigma_2 + c\sigma_3) < 0$ and which reduces smoothly to $s_0$ as $r \to 0$,

- the 'scaling' branch $s_{\mathrm{sc}}(r)$, which exists whenever the sign of $r(a\sigma_1 + b\sigma_2 + c\sigma_3) < 0$, and is such that the distances $|\Sigma_a - \Sigma_b|$ are of order $r$ as $r \to 0$.

To establish this claim, we may proceed as in [12, §3.1]. Let us define

$$
z_1 = \frac{a}{|\Sigma_1 - \Sigma_2 - r|}, \quad z_2 = \frac{b}{|\Sigma_2 - \Sigma_3 - r|}, \quad z_3 = \frac{c}{|\Sigma_3 - \Sigma_1 - r|}.
\tag{3.38}
$$

One may eliminate $z_1, z_2$ in favor of $z_3$ using (3.36) to in terms of $z_3$, and fix the latter through the requirement $(\Sigma_1 - \Sigma_2 - r) + (\Sigma_2 - \Sigma_3 - r) + (\Sigma_3 - \Sigma_1 - r) = -3r$. Thus, solutions to (3.36) are equivalent to solutions of $f(z_3) = -3r$ for $z_3 \in \mathbb{R}^+$, where $f$ is the function defined in [12, (3.7)].

$$
f(z_3) = \frac{a\sigma_1}{z_3 + 2\zeta_1} + \frac{b\sigma_2}{z_3 - 2\zeta_3} + \frac{c\sigma_3}{z_3}.
\tag{3.39}
$$

For $\zeta_1 > 0, \zeta_3 < 0$, $f$ is regular in $]0, +\infty[$, blows up at 0 and decays at $\infty$ according to

$$
f(z_3) \sim \frac{c\sigma_3}{z_3} \ (z_3 \to 0^+), \qquad f(z_3) \sim \frac{a\sigma_1 + b\sigma_2 + c\sigma_3}{z_3} \ (z_3 \to +\infty).
\tag{3.40}
$$

For small (negative) $r$ and $a\sigma_1 + b\sigma_2 + c\sigma_3 > 0$, there is always a solution to $f(z_3) = -3r$ at large $z_3$, given by $z_3 = -(a\sigma_1 + b\sigma_2 + c\sigma_3)/r$. This corresponds to the scaling branch, obtained by $\zeta_a = 0$ in (3.32),

$$
\Sigma_1 - \Sigma_2 - r = -\frac{3a\sigma_1}{a\sigma_1 + b\sigma_2 + c\sigma_3}r + \mathcal{O}(r^2\zeta),
\tag{3.41}
$$

and cyclic permutations thereof, where the corrections can be computed systematically as a Taylor series in $r\zeta$.

On the other hand, when the signs of $\sigma_3$ and $a\sigma_1 + b\sigma_2 + c\sigma_3$ are opposite, it is clear that for small enough $r$, there must exist at least one solution in the range $]0, +\infty[$. In the $r = 0$ case discussed in [12], it was argued that multiple regular solutions cancel in pair in (3.33), since sgn($\det \partial_a \partial_b \mathcal{W}$) will be opposite. This no longer true in the present case, since they may lead to different contour integrals in (3.34). In particular, when $r$ is small (and negative), $a\sigma_1 + b\sigma_2 + c\sigma_3 > 0$ and $\sigma_3 < 0$, there are two solutions, the scaling solution $z_3 = -(a\sigma_1 + b\sigma_2 + c\sigma_3)/r$ and one of which reduces to the usual solution $s_0$ as $r \to 0$.

After considering the various sign choices, we find that when $a, b, c$ satisfy the triangular inequalities, there are two regular solutions (with signs $\sigma_{1,2,3} = + + -$ and $- - +$) and four scaling solutions (with signs $+ + +$, $+ - +$, $- + +$, $+ + -$). In contrast, if one of the triangular inequalities is violated, say $c > a + b$, then there are four scaling solutions, with the same signs $+ + +$, $+ - +$, $- + +$, $- - +$.

In Appendix C, we compute the contour integral (3.34) using residue calculus. After combining the contributions from the 4 possible sign choices, we find that

$$\mathcal{I}_{\text{sc}} = \Omega_S(\gamma, y) + H(\{\gamma_1, \gamma_2, \gamma_3\}, y), \tag{3.42}$$

such that the total index reproduces the standard result from the Coulomb branch formula,

$$\mathcal{I} = [g_C(\{\gamma_1, \gamma_2, \gamma_3\}) + H(\{\gamma_1, \gamma_2, \gamma_3\}, y)]\overline{\Omega}(\gamma_1, y)\overline{\Omega}(\gamma_2, y)\overline{\Omega}(\gamma_3, y) + \Omega_S(\gamma, y). \tag{3.43}$$

It is worth noting that $\mathcal{I}$ vanishes in the case where one of the triangle inequalities $a < b + c$, $b < c + a$, $c < a + b$ is violated, even though the contributions of the four collinear scaling scolutions are separately non-zero. In fact, $\mathcal{I}$ vanishes unless the quantum version (2.14) of these inequalities holds.[9] In §4.5, we extend these results to arbitrary Abelian cyclic quivers, using a more efficient method for evaluating the the contour integral (3.34).

### 3.4.2 Abelian quiver with multiple oriented cycles

In the presence of multiple oriented cycles, in addition to the regular collinear solutions there can exist different types of scaling solutions where a subset $C \subset Q_0$ of the centers (which we call a cluster) can become arbitrarily close. As observed in [34], a necessary condition is that the subquiver $Q'$ corresponding to this cluster cannot be decomposed into a disjoint union $Q'_A \cup Q'_B$ where arrows go only from $A$ to $B$ or $B$ to $A$. This condition is equivalent to requiring that $Q'$ is strongly connected.[10] In particular, $Q'$ must itself have oriented loops.

The distinct scaling regimes therefore correspond to lists of clusters $\{C_P^{(i)}, i = 1 \ldots M_P\}$ such that the associated quivers $Q_P^{(i)}$ are strongly connected, and $P \in S$ labels the possible such configurations. Upon collapsing all the nodes in each cluster into a single node, we obtain an Abelian quiver $Q_P^{(0)}$ with $M_P$ nodes and adjacency matrix $\alpha_{ij} = \sum_{a \in C_P^{(i)}} \sum_{b \in C_P^{(j)}} \kappa_{ab}$, which need not be strongly connected. Accordingly, the total Witten index decompose as $\mathcal{I} = \sum_{P \in S} \mathcal{I}_P$. Note that the trivial configuration $P_0$ where all clusters contain a single element corresponds to the contribution of regular collinear solutions, $\mathcal{I}_{P_0} = \mathcal{I}_{\text{reg}}$.

Now, for each node $a$ in the cluster $C_P^{(i)}$, we separate both $u_a$ and $D_a$ into a 'center of motion' and a 'fluctuation' part

$$u_a = u_0^{(i)} + \epsilon_a, \quad D_a = D_0^{(i)} + \delta_a, \quad \text{with} \quad \sum_{a \in C_P^{(i)}} \epsilon_a = \sum_{a \in C_P^{(i)}} \delta_a = 0. \tag{3.44}$$

---

[9]From the Higgs branch point of view, these conditions are obvious since the expected dimension of the Higgs branch is $a + b - c - 2$, or permutations thereof depending on the chamber. From the Coulomb branch point of view, $\Omega_S$ vanishes and $g_C$ is cancelled by $H$.

[10]Recall that a graph is strongly connected if every vertex is reachable from every other vertex.

In the limit $\beta \to \infty$, $\epsilon_a$ is much smaller than differences between $u_0^{(i)}$'s, and similarly for $\delta_a$, it is then straightforward to show that the product of determinants (3.4) factorizes into

$$g(u,D) = g_P^{(0)}(u_0, D_0) \prod_{i=1}^{M_P} g_P^{(i)}(\epsilon, \delta), \tag{3.45}$$

where $g_P^{(i)}(u_i, D_i)$ is the one-loop determinant associated to $Q_P^{(i)}$. Similarly, the 'classical action' (3.20) decomposes into

$$S(D, \zeta) = S_P^{(0)}(D_0, \zeta_P) + \sum_{i=1}^{M_P} S_P^{(i)}(\delta, 0), \tag{3.46}$$

where $S_P^{(i)}(D, \zeta)$ is the action (3.20) for the quiver $Q_P^{(i)}$ and the FI parameters $\zeta_P$ are equal to $\sum_{a \in C_P^{(i)}} \zeta_a$. Finally, the measure decomposes as

$$\det h(u,D)\, du\, d\bar{u} = \det h_P^{(0)}(u_0, D_0)\, du_0\, d\bar{u}_0 \prod_{i=1}^{M_P} \det h_P^{(i)}(\epsilon, \delta)\, d\epsilon\, d\bar{\epsilon}, \tag{3.47}$$

where $h_P^{(i)}$ the matrix (3.7) for the quiver $Q_P^{(i)}$. We conclude that the integral $\mathcal{I}_P$ factorizes into

$$\mathcal{I}_P = \mathcal{I}_{\text{reg}}(Q_P^{(0)}, \zeta_P, y) \times \prod_{i=1}^{M_P} \mathcal{I}_{\text{sc}}^{\max}(Q_P^{(i)}), \tag{3.48}$$

where $\mathcal{I}_{\text{reg}}(Q_P^{(0)}, \zeta_P, y)$ is the regular part of the Witten index of the quiver $Q_P^{(0)}$ with FI parameters $\zeta_P$ and $\mathcal{I}_{\text{sc}}^{\max}(Q_P^{(i)})$ is the contribution to the index of the sub-quiver $Q_P^{(i)}$ from 'maximally scaling solutions' where all centers collide (note that the index of $Q_P^{(i)}$ may also receive contributions from scaling solutions where only a subset of the centers collide). This is indeed of the form predicted by the Coulomb branch formula (2.18), upon identifying $\mathcal{I}(Q_P^{(0)}, \zeta_P, y)$ with the Coulomb index $g_C(\{\alpha_{ij}, c_i\}, y)$ and $\mathcal{I}_{\text{sc}}^{\max}(Q_P^{(i)})$ with the 'total invariant' $\Omega_{\text{T}}(\alpha_i)$. The relation (2.19) can be viewed as a recursive definition of the single-centered invariants, which unlike $\Omega_{\text{T}}(\alpha_i)$ are bona fide symmetric Laurent polynomials.

## 3.5 Non-Abelian quivers

In this section, we briefly discuss the case of a general dimension vector $\gamma = (N_1, \dots, N_K)$, restricting to quivers without oriented cycles for simplicity. The fact that $N_a > 1$ leads to two complications: first, the vector multiplet determinant (3.5) is no longer independent of the Cartan variables, due to the contributions of roots of $U(N)$ with $s \neq s'$; and second, we can no longer assume that $\beta|\Sigma_{ss'}| \gg 1$ in the chiral multiplet determinant (3.14), since there is no potential preventing the Cartan variables within one $U(N)$ factor to coincide. Nonetheless, we shall argue that the problem separates into a product of $SU(m)$ non-Abelian dynamics associated to $m$ nearly coincident Cartan variables, which can be treated using the usual Jeffrey-Kirwan residue prescription, and the Abelian dynamics of the center of motion in each $SU(m)$ factor, which can be treated as in the previous section. One way of separating these variables is to apply the Cauchy-Bose identity for each of the $U(N_a)$ vector multiplet determinants, as explained in [35], and then recombine the corresponding sum over permutations into a product of $U(m)$ determinants. However, it is more economical to proceed as follows, similarly to the case of of Abelian quivers with multiple cycles in §3.4.2.

Consider all possible partitions $P$ of $N_a = \sum_{k=1}^{M_a} m_a^k$ for each $a = 1, \dots, K$. The partition $P$ splits the Cartan variables $\{u_\alpha, \alpha = 1 \dots r\} = \{u_{a,s}\}$ into clusters $\{u_\alpha, \alpha \in P_{a,k}\}$ of size $m_a^k$.

We shall decompose the domain of integration in (2.7) into regions $\mathcal{M}_P$ where the differences $|\Sigma_\alpha - \Sigma_{\alpha'}|$ with $\Sigma_\alpha = -\frac{2\pi}{\beta}\text{Im}\,u_\alpha$ are greater than $1/\beta$ whenever $\alpha$ and $\alpha'$ belong to different clusters, and smaller than $1/\beta$ when they are in the same cluster. Thus the integral (3.2) decomposes as $\mathcal{I} = \sum_P \mathcal{I}_P$.

Now, for a given partition $P$, in each cluster we separate both $u_\beta$ and $D_\beta$ into a 'center of motion' and 'fluctuation' part

$$u_\alpha = u_0^{(a,k)} + \epsilon_\alpha, \quad D_\alpha = D_0^{(a,k)} + \delta_\alpha, \quad \text{with} \quad \sum_{\alpha \in P_{a,k}} \epsilon_\alpha = \sum_{\alpha \in P_{a,k}} \delta_\alpha = 0. \tag{3.49}$$

In the limit $\beta \to \infty$, $\epsilon_\alpha$ is much smaller than differences between $u_0^{(a,k)}$'s, and similarly for $D_\beta$. It is then straightforward to show that the product of determinants (3.4) factorizes into

$$g(u,D) = g_{\text{abelian}}(u_0, D_0) \prod_{a=1}^{K} \prod_{k=1}^{M_a} \tilde{g}_{\text{vector}}^{(m_a^k)}(\epsilon, \delta), \tag{3.50}$$

where $g_{\text{abelian}}(u_0, D_0)$ is the one-loop determinant associated to an Abelian quiver $Q_P$ with $\sum_{a=1}^{K} M_a$ nodes and adjacency matrix $\kappa_{(a,k)(b,k')} = m_a^k m_b^{k'} \kappa_{ab}$, while $\tilde{g}_{\text{vector}}^{(m)}$ is equal to $g_{\text{vector}}^{(m)}$ in (3.5) multiplied by $\sin \pi z$). Similarly, the 'classical action' (3.20) decomposes into

$$S(D, \zeta) = S_{\text{abelian}}(D_0, \zeta_P) + \frac{1}{2e^2} \sum_{a=1}^{K} \sum_{k=1}^{M_a} \sum_{\alpha \in P_{a,k}} \epsilon_\alpha^2, \tag{3.51}$$

where the FI parameters $\zeta_P$ are equal to $m_a^k \zeta_a$, and the measure as

$$\det h(u, D)\, du\, d\bar{u} = \det h_{\text{abelian}}(u_0, D_0)\, du_0\, d\bar{u}_0 \prod_{a=1}^{K} \prod_{k=1}^{M_a} \det h_{SU(m_a^k)}(\epsilon, \delta)\, d\epsilon\, d\bar{\epsilon}, \tag{3.52}$$

where the $(m-1) \times (m-1)$ matrix $h_{SU(m)}$ is defined similarly to the first term of (3.7). We conclude that the integral $\mathcal{I}_P$ factorizes into

$$\mathcal{I}_P = \frac{\prod_{a=1}^{K} \prod_{k=1}^{M_a} m_a^k!}{\prod_{a=1}^{K} N_a!} \mathcal{I}_{Q_P} \times \prod_{a=1}^{K} \prod_{k=1}^{M_a} \bar{\Omega}_{SU(m_a^k)}, \tag{3.53}$$

where $\mathcal{I}_{Q_P}$ is the index of the quiver $Q_P$ with FI parameters $\zeta_P$, $\bar{\Omega}_{SU(m)}$ is the index of QQM with a single node of rank $m$, and the prefactor in (3.53) comes from the factors of $1/|W|$ in the integral representations of $\mathcal{I}_P$ and $\bar{\Omega}_{SU(m)}$. The latter was first computed in [36], and rederived in the Jeffrey-Kirwan formalism in [37]

$$\bar{\Omega}_{SU(m)} = \frac{y - 1/y}{m(y^m - y^{-m})}. \tag{3.54}$$

This is recognized as the rational index $\bar{\Omega}_S(\alpha_i)$ for a dimension vector $\alpha_i$ to equal $m$ times a basic dimension vector $(0, \ldots, 1, 0, \ldots)$ associated to the $i$-th node of $Q_P$ (for any $i$). After collecting all partitions with the same shape, the total index can therefore be written as

$$\mathcal{I} = \sum_{\gamma = \sum_{i=1}^{n} \alpha_i} \frac{\Omega_{Q(\{\alpha_i\})}}{|\text{Aut}\{\alpha_i\}|} \prod_{i=1}^{n} \bar{\Omega}_S(\alpha_i), \tag{3.55}$$

where $Q(\{\alpha_i\})$ is the Abelian quiver $Q_P$ defined above, $|\text{Aut}\{\alpha_i\}|$ is the order of the subgroup of permutations of $n$ elements which preserve the ordered list $\{\vec{N}_i, i = 1 \ldots n\}$ and $\bar{\Omega}(\alpha_i)$ equals

to (3.54) whenever $\alpha_i$ is $m$ times a basic vector, or 0 otherwise. Thus, we have reproduced the Coulomb formula (2.15). for non-Abelian quivers without oriented cycles.

Unfortunately, the derivation above remains heuristic, since we have not justified the implicit assumption that all contributions originate from a region where $\epsilon_\alpha \ll u_0$. It would also be interesting to extend these arguments to non-Abelian quivers with oriented cycles, and elucidate the origin of the 'partial' single-centered invariants' introduced in our previous work [35, §5.4].

# 4 Cyclic quivers with generic superpotential

Our goal in this section is to compute the scaling index (3.34) for cyclic quivers with an arbitrary number $K$ of nodes, and $a_\ell \geq 1$ arrows from $\ell$ to $\ell + 1$ with the node $K + 1$ identified with the first node. Before doing so however, we shall expand on some known results for the single-centered and attractor invariants for such quivers, derive the so-called stacky invariants for trivial stability (which turn out to have an interesting connection to the scaling invariant), and also comment on an intriguing connection to the combinatorics of derangements. The hurried reader only interested in the scaling index may skip ahead to §4.5.

In order to state the indices associated to all such quivers at once, it is useful to define the generating series, e.g. for single-centered indices

$$Z_S(\{x_1, \ldots, x_K\}, y) = \sum_{a=1}^{\infty} \cdots \sum_{a_K=1}^{\infty} \Omega_S(\{a_1, \ldots, a_\ell\}, y) \, x_1^{a_1} \ldots x_K^{a_K}. \qquad (4.1)$$

Moreover, we denote by $\tau_\ell$ the elementary symmetric functions in the $x_\ell$'s, namely $\tau_1 = \sum x_\ell$, $\tau_2 = \sum_{\ell < m} x_\ell x_m$, etc.

## 4.1 Single-centered indices

The generating functions of single-centered indices $\Omega_S(\{a_\ell\}, 1)$ was computed in [12, Eq. (4.29)]:

$$Z_S(\{x_\ell\}, y) \quad := \quad \frac{1}{2} \frac{(y - 1/y)\left[y \prod_{\ell=1}^{K} \frac{x_\ell}{1 + y x_\ell} + y^{-1} \prod_{\ell=1}^{K} \frac{x_\ell}{1 + x_\ell/y}\right]}{y \prod_{\ell=1}^{K}(1 + x_\ell/y) - y^{-1} \prod_{\ell=1}^{K}(1 + y x_\ell)} \qquad (4.2)$$

$$+ \frac{1}{2} \sum_{k=1}^{K} \frac{1 - x_k^2}{(1 + y x_k)(1 + x_k/y)} \prod_{\substack{\ell=1\ldots K \\ \ell \neq k}} \frac{x_\ell}{(1 - x_\ell/x_k)(1 - x_\ell x_k)}.$$

While this formula is manifestly invariant under $y \to 1/y$ and under permutations of the $x_i$'s, it is rather unwieldy, since it appears to have poles at $x_\ell = y$ and $x_\ell = 1/y$, while the single-centered indices must be Laurent polynomials in $y$. In Appendix B, we show that this formula can be rewritten as

$$Z_S(\{x_\ell\}, y) \quad = \quad \prod_{\ell=1}^{K} \frac{x_\ell}{(1 + x_\ell/y)} \left[\frac{1}{y \, \Delta_K(\{x_\ell\}, y)} + \frac{P_K(\{x_\ell\}, -1/y)}{\prod_{1 \leq k < \ell \leq K}(1 - x_k x_\ell)}\right], \qquad (4.3)$$

where $\Delta_K$ is the symmetric polynomial

$$\Delta(\{x_\ell\}, y) \quad = \quad \frac{y \prod_{\ell=1}^{K}(1 + x_\ell/y) - y^{-1} \prod_{\ell=1}^{K}(1 + y x_\ell)}{y - 1/y} = 1 - \sum_{\ell=1}^{K-1} \kappa(\ell) \tau_{\ell+1}, \qquad (4.4)$$

where $\kappa(\ell) = \frac{y^\ell - y^{-\ell}}{y - 1/y}$, while $P_K$ is a symmetric polynomial satisfying the recursion

$$P_K(\{x_1, \ldots, x_K\}, w) = \frac{w}{w - x_K} \tag{4.5}$$

$$\times \left[ P_{K-1}(\{x_1, \ldots, x_{K-1}\}, w) \prod_{\ell=1}^{K-1} (1 - x_\ell x_K) - P_{K-1}(\{x_1, \ldots, x_{K-1}\}, x_K) \prod_{\ell=1}^{K-1} (1 - w x_\ell) \right],$$

with $P_2(\{x_1, x_2\}, w) = w$. Just like $\Delta_K$, the polynomial $P_K$ is independent of $K$, given by

$$
\begin{aligned}
P_K(\{x_\ell\}, w) &= w\Big[ 1 - w\tau_3 + \tau_4(w^2 - 1 + w\tau_1 - w^2\tau_4) \\
&\quad + \tau_5\big( -w^3 + (1 - w^2)\tau_1 - w\tau_1^2 + w\tau_2 + w^2\tau_3 + w(w^2 - 1)\tau_4 + w^2\tau_1\tau_4 \\
&\quad + \tau_5(-1 + w(1 - w^2)\tau_1 - w^2\tau_2 + w^2\tau_5)\big) + \ldots \Big],
\end{aligned}
\tag{4.6}
$$

where the dots vanish when $K \leq 5$.

While the two terms in (4.3) separately have poles at $x_\ell = -y$, their sum does not, thanks to the property

$$x_k P_K(\{x_\ell\}, 1/x_k) = \prod_{\substack{1 \leq m < \ell \leq K \\ m \neq k, \ell \neq K}} (1 - x_m x_\ell). \tag{4.7}$$

Moreover, the sum is in fact invariant under $y \to 1/y$, due to the identity

$$P_K(\{x_1, \ldots, x_K\}, w) \prod_{\ell=1}^{K} (1 - x_\ell/w) - P_K(\{x_1, \ldots, x_K\}, 1/w) \prod_{\ell=1}^{K} (1 - x_\ell w)$$

$$= (w - 1/w) \prod_{1 \leq m < \ell \leq K} (1 - x_m x_\ell), \tag{4.8}$$

which follows from (4.7) after setting $w = 1/x_K$. In fact, one can rewrite (4.3) in a form that makes both properties manifest:

$$Z_S(\{x_\ell\}, y) = \frac{N_K(\{x_\ell\}, y) \prod_{\ell=1}^{K} x_\ell}{\Delta_K(\{x_\ell\}, y) \prod_{k < \ell} (1 - x_k x_\ell)}, \tag{4.9}$$

where $N_K$ is again a universal symmetric polynomial, given for $K \leq 5$ by

$$
\begin{aligned}
N_K(\{x_\ell\}, y) &= \tau_3 - \tau_1\tau_4 + \tau_5(\tau_1^2 - \tau_2 + \tau_4) - \tau_1\tau_5^2 \\
&\quad + (y + 1/y)\left( \tau_4 - \tau_4^2 - \tau_1\tau_5 + \tau_3\tau_5 + \tau_1\tau_4\tau_5 - \tau_2\tau_5^2 \right) \\
&\quad + (y^2 + 1 + 1/y^2)\left( \tau_5 - \tau_4\tau_5 + \tau_1\tau_5^2 - \tau_5^3 \right) + \ldots.
\end{aligned}
\tag{4.10}
$$

The leading asymptotic growth of $\Omega(\{a_\ell\}, y)$ as $a_\ell \to \infty$ is governed by the factor $\Delta_K$ in the denominator. In the special case where all $a_\ell$ are all equal to $a$, the unrefined single-centered index can be shown to grow as [12].

$$\Omega_S(\{a_\ell\}, 1) \sim a^{\frac{1-K}{2}} (K - 1)^{Ka}. \tag{4.11}$$

## 4.2 Attractor indices

The attractor indices $\Omega_\star(\gamma, y)$ are defined as the Witten index $\Omega(\gamma, y, \zeta^\star)$ for special value of the FI parameters $\zeta_a^\star = -\kappa_{ab} N_b$, where $\kappa_{ab}$ is the skew-symmetric adjacency matrix and $N_a$ is the dimension vector [18,34]; in particular, they satisfy $\sum_a N_a \zeta_a^\star = 0$. They differ from single-centered indices $\Omega_S(\gamma)$ precisely due to multicentered solutions which have scaling regions.

For cyclic quivers, the refined index $\Omega(\gamma, y, \zeta)$ was computed in [12, (4.21)] in a particular chamber $\mathcal{C}$ where $\zeta_i > 0$ for $i = 1 \ldots K-1$, $\zeta_K < 0$. After subtracting the contributions with $a_K = 0$, one obtains[11]

$$
\begin{aligned}
Z_{\mathcal{C}} &= \left[ \frac{y \prod_{\ell=1}^{K-1}(1 + x_\ell/y) - y^{-1} \prod_{\ell=1}^{K-1}(1 + y x_\ell)}{y \prod_{\ell=1}^{K}(1 + x_\ell/y) - y^{-1} \prod_{\ell=1}^{K}(1 + y x_\ell)} - 1 \right] \prod_{\ell=1}^{K-1} \frac{x_\ell}{(1 + x_\ell y)(1 + x_\ell/y)} \\
&= \left[ \frac{\sum_{\ell=1}^{K} \kappa(\ell)\tau_\ell}{1 - \sum_{\ell=1}^{K-1} \kappa(\ell)\tau_{\ell+1}} - x_K \right] \prod_{\ell=1}^{K} \frac{x_\ell}{(1 + x_\ell y)(1 + x_\ell/y)},
\end{aligned} \tag{4.12}
$$

where we used the same notation $\kappa(\ell)$ as in (4.4). In the unrefined limit, this reduces to

$$
Z_{\mathcal{C}}(1) = -\frac{x_K}{1 + x_K} \prod_{\ell=1}^{K-1} \frac{x_\ell}{(1 + x_\ell)^2} + \left( 1 - \sum_{\ell=1}^{K} \frac{x_\ell}{1 + x_\ell} \right)^{-1} \prod_{\ell=1}^{K} \frac{x_\ell}{(1 + x_\ell)^2}. \tag{4.13}
$$

The chamber $\mathcal{C}$ coincides with the attractor chamber when $a_K$ is the largest of all $a_\ell$'s Other cases can of course be gotten by permuting the $x_\ell$'s. It is thus straightforward in principle to construct the generating series of attractor indices $Z_\star$. In practice however, it is complicated and we have computed it only for $K = 3$ and 4. For $K = 3$, we find

$$
Z_\star = \frac{x_1^2 x_2^2 x_3^2 \left( 2 - x_1 x_2 - x_2 x_3 - x_3 x_1 + x_1^2 x_2^2 x_3^2 \right)}{(1 - x_1 x_2)(1 - x_2 x_3)(1 - x_3 x_1)(1 + x_1 x_2 x_3)^2 (1 - \tau_2 - (y + 1/y)\tau_3)}. \tag{4.14}
$$

The result for $K = 4$ can be found in Appendix B.3. The main point is that the generating series is symmetric under permutations of the $x_\ell$'s, and has a factor of $\Delta_K(y)$ in the denominator, which implies the same exponential growth as (4.11) when $a_\ell \to \infty$. The difference between $Z_\star$ and $Z_S$ is free from this factor, in agreement with the fact that it stems from scaling solutions, whose index grows only polynomially, e.g. for $K = 3$

$$
Z_\star - Z_S = \frac{\tau_3^2}{(1 - x_1 x_2)(1 - x_2 x_3)(1 - x_3 x_1)(1 + y\tau_3)(1 + \tau_3/y)}. \tag{4.15}
$$

It is easy to check that the Taylor coefficients are non-vanishing only when $a, b, c$ satisfy the triangular inequalities.

## 4.3 Invariants for trivial stability

For later purposes, it will be useful to compute yet a different set of chamber-independent indices associated to cyclic quivers: namely the stacky invariants for trivial stability condition $\zeta = 0$. While their physical meaning is a priori obscure (see however the next section), they are mathematically well defined, and computable from the DT invariants in any given chamber by the Reineke formula (see Appendix B).

Recall that stacky invariants are related to the rational DT invariants by [27]

$$
\mathcal{A}(\gamma; w) = \sum_{\substack{\sum_{i=1}^{k} \alpha_i = \gamma, \\ \mu(\alpha_i) = \mu(\gamma)}} \frac{1}{k!} \prod_{i=1}^{k} \left( \frac{\overline{\Omega}(\alpha_i; -w^{-1})}{w - w^{-1}} \right), \tag{4.16}
$$

---

[11]For $K = 2$, this reduces to $Z_{\mathcal{C}} = x_1^2 x_2 / [(1 + x_1 y)(1 + x_1/y)(1 - x_1 x_2)]$, whose Taylor coefficients vanish for $a_1 < b_1$ and are given by the Poincaré-Laurent polynomial of the projective space $\mathbb{P}^{a_1 - a_2 - 1}$ for $a_1 \geq b_1$. This is in agreement with the fact the Higgs branch is the intersection of $a_2$ hyperplanes inside $\mathbb{P}^{a_1 - 1}$. In contrast, the series $Z_S$ for $K = 2$ vanishes since $\tau_k = 0$ for $k > 2$.

where $\mu(\beta)$ is the 'slope' of the dimension vector $\beta = \sum_a n_a \gamma_a$, defined by

$$\mu(\beta) \equiv \frac{\sum_a \zeta_a n_a}{\sum_a n_a}. \tag{4.17}$$

Recall that the parameters $\zeta_\ell$ are chosen to satisfy $\sum_a \zeta_a N_a = 0$, hence $\mu(\gamma) = 0$. For generic stability parameters and dimension vector such that $N_a \leq 1$, Eq. (4.16) reduces to

$$\mathcal{A}(\gamma; w) = \frac{\overline{\Omega}(\gamma; -w^{-1})}{w - w^{-1}}, \tag{4.18}$$

hence $\mathcal{A}(\gamma; w)$ contains the same information as $\overline{\Omega}(\gamma, y, \zeta)$ with $w = -1/y$. For vanishing superpotential, the stacky invariant at trivial stability is easily calculated, e.g. by counting representations over finite fields,

$$\mathcal{A}_0^{W=0}(\gamma; w) = \frac{w^{\sum_{a:\ell \to k} N_\ell N_k - \sum_\ell N_\ell^2}}{\prod_{\ell=1}^{K} \prod_{j=1}^{N_\ell} (1 - w^{-2j})}. \tag{4.19}$$

In our case however, the superpotential is non-trivial and (4.19) does not apply. Instead, by using Reineke's formula we show in Appendix B that the generating series of stacky invariants $\mathcal{A}_0(\gamma; w)$ for trivial stability condition is given by

$$Z_{\mathcal{A}_0} = Z_{\mathcal{A}_0^{W=0}} - \frac{w}{(w - 1/w)\Delta_K(-1/w)} \prod_{\ell=1}^{K} \frac{x_\ell}{(1 - wx_\ell)}, \tag{4.20}$$

where $Z_{\mathcal{A}_0^{W=0}}$ is the generating series of the invariants (4.19),

$$Z_{\mathcal{A}_0^{W=0}} = \left(\frac{w}{w - 1/w}\right)^K \prod_{\ell=1}^{K} \frac{x_\ell}{(1 - wx_\ell)}. \tag{4.21}$$

As for the generating series of single-centered indices and attractor indices, (4.20) is symmetric under permutations of $x_i$'s and exhibits the conspicuous factor $1/\Delta_K(y)$ which is responsible for the exponential growth as $a_\ell \to \infty$. In contrast to the previous ones however, $Z_{\mathcal{A}_0}$ is not invariant under $y \to 1/y$. Defining the generating series of 'trivial stability indices'

$$Z_{\text{triv}}(y) = (y - 1/y) Z_{\mathcal{A}_0}(-1/y), \tag{4.22}$$

we find that the difference between trivial stability and single-centered indices is given by

$$Z_S - Z_{\text{triv}} = \prod_{\ell=1}^{K} \frac{x_\ell}{(1 + x_\ell/y)} \left[ -\frac{y^{-K}}{(y - 1/y)^{K-1}} + \frac{P_K(\{x_\ell\}; -1/y)}{\prod_{1 \leq k < \ell \leq K} (1 - x_k x_\ell)} \right]. \tag{4.23}$$

## 4.4 Connections to derangements

A common in all generating series discussed above is the factor $\Delta_K(\{x_\ell\}, y)$ in the denominator, where $\Delta_K$ is the symmetric polynomial in (4.4). As noted in [15, §3.5] in the case $K = 3$, and in [12, §4.3] for any $K$, the inverse of this factor $1/\Delta_K(\{x_\ell\}, 1)$ in the unrefined limit has a simple combinatorial meaning: it is the generating series of the number of derangements $D(\{a_\ell\})$ of a set of $N = \sum_\ell a_\ell$ objects consisting of $a_1$ objects of color 1, $a_2$ objects of color 2, etc. Here, we generalize this observation to the refined case.

Recall that derangements of a set are permutations $\sigma \in S_N$ such that no object of color $c(i)$ ends up in a slot formerly occupied by an object of the same color: $c(\sigma((i)) \neq c(i)$ for

all $i = 1 \dots N$. To see that $1/\Delta_K(1)$ is the generating series of the number of derangements $D(\{a_\ell\})$ [38, 39],[12] notice that $\Delta_K(\{x_\ell\}, 1)$ can be written as a $K \times K$ determinant

$$\Delta_K(\{x_\ell\}, 1) = \det[\delta_{ij} - a_{ij} x_j], \tag{4.24}$$

where $a_{ij} = 1$ if $i \neq j$ or 0 if $i = j$. On the other hand, MacMahon's 'master formula' states that the coefficient of $x_1^{a_1} \dots x_K^{a_K}$ in the Taylor expansion of the inverse of the above determinant is equal to the coefficient of the same term in the expansion of

$$(a_{11} x_1 + \dots + a_{1K} x_K)^{a_1} (a_{21} x_1 + \dots + a_{2K} x_K)^{a_2} \dots (a_{K1} x_1 + \dots + a_{KK} x_K)^{a_K}. \tag{4.25}$$

Equivalently, the inverse determinant can be interpreted as the generating series of (possibly disconnected) closed circuits on a quiver with $K$ nodes and arrows between each pair of nodes, where each edge $i \to j$ in the circuit is weighted by $a_{ij} x_j$. For $a_{ij} = 0$ when $i = j$, circuits with fixed points cancel out and thus correspond to derangements, counted with unit weight when $a_{ij} = 1$ for $i \neq j$. This combinatorial interpretation makes it clear that the Taylor coefficients of the generating series of single-centered and attractor indices vanish unless the $a_\ell$'s satisfy the polygonal inequalities (2.13).[13]

In order to interpret the factor $1/\Delta_K(\{x_\ell\}, y)$ appearing in the generating series of refined single-centered indices, it suffices to note that it can be rewritten in the same form as (4.24), where $a_{ij} = y$ if $i > j$, $1/y$ if $i < j$ or 0 if $i = j$ [38]. It follows that the Taylor coefficients of $1/\Delta_K(y)$ are given by

$$D(\{a_\ell\}, y) = \sum_\sigma y^{n_+(\sigma) - n_-(\sigma)}, \tag{4.26}$$

where $\sigma$ runs over derangements of $N = \sum_{\ell=1}^K a_\ell$ colored objects, $n_+(\sigma)$ is the number of $i$ such that $c(\sigma(i)) > c(i)$ and $n_-(\sigma)$ is the number of $i$ such that $c(\sigma(i)) < c(i)$ (note that $c(\sigma(i)) \neq c(i)$ by the derangement condition). Here we use the standard coloring, with $c(i) = 1$ for $i = 1 \dots a_1$, $c(i) = 2$ for $i = a_1 + 1 \dots a_1 + a_2$, etc, up to $c(N) = K$. For example, when all $a_i$'s are set to 1, corresponding to derangements of distinct objects, one finds

$$
\begin{aligned}
D_1 &= 0, \quad D_2 = 1, \quad D_3 = \kappa(2) = y + 1/y, \\
D_4 &= \kappa(3) + 6\kappa(1)^2 = y^2 + 7 + 1/y^2 \\
D_5 &= \kappa(4) + 20\kappa(1)\kappa(2) = y^3 + 21y + 21/y + 1/y^3, \dots \\
D_6 &= \kappa(5) + 30\kappa(1)\kappa(3) + 20\kappa(2)^2 + 90\kappa(1)^3 = y^4 + 51y^2 + 161 + 51/y^2 + 1/y^4
\end{aligned}
\tag{4.27}
$$

where $\kappa(m) = (y^m - y^{-m})/(y - 1/y) = y^{m-1} + y^{m-3} + \dots + y^{1-m}$. Note this refinement of the number of derangements differs from the one considered in [40], which (unlike the present one, to our knowledge) admits a simple $q$-deformed version of the classic recursion formulae

$$D_K = (K-1)(D_{K-1} + D_{K-2}) \quad \Rightarrow \quad D_K = K D_{K-1} + (-1)^K. \tag{4.28}$$

## 4.5 Scaling solutions for Abelian cyclic quivers

Let us consider the case of a cyclic quiver with $K$ nodes and $\kappa_{\ell, \ell+1} = a_\ell > 0$ arrows from vertex $\ell$ to vertex $\ell + 1$, with $\ell \in \mathbb{Z}/K\mathbb{Z}$. The equations (3.32) reduce to

$$\frac{a_\ell}{|\Sigma_{\ell, \ell+1}|} - \frac{a_{\ell-1}}{|\Sigma_{\ell-1, \ell}|} = 2\zeta_\ell, \quad \ell = 1, \dots, K. \tag{4.29}$$

---

[12]We are grateful to P. di Francesco and J-B. Zuber for discussions on derangements, and to M. Ismaïl for bringing the important reference [38] to our attention after the first release of this work.

[13]Just as in the $K = 3$ case [15], the additional factors of $(1 - x_k x_\ell)$ appearing in (4.9) imply that they in fact vanish unless the strong constraints (2.14) are satisfied.

The existence of solutions can be analyzed using the same method as in §3.4.1, and amounts to deforming the equation $f(z_K) = 0$ in [12, §4.1] into $f(z_K) = -\frac{2\pi}{\beta}\mathrm{Im}z$. We focus on the scaling branch in the deep scaling regime $\zeta_\ell = 0$. The solution is then given by

$$|\Sigma^\star_{\ell,\ell+1}| = -\frac{2\pi\mathrm{Im}z}{\beta}\frac{a_\ell}{\sum_{\ell=1}^{K}\sigma_\ell a_\ell}, \tag{4.30}$$

with[14] $\sigma_\ell = \mathrm{sgn}\Sigma^\star_{\ell,\ell_i}$. Since the left hand side of (4.30) is positive, the signs $\sigma_i$ must be chosen such that $\mathrm{sgn}(\sum_{\ell=1}^{n}\sigma_\ell a_\ell) = -\mathrm{sgn}\,\mathrm{Im}z$, which selects $2^{n-1}$ out of total $2^n$ possible choices of sign contribute. For definiteness, we shall choose $\mathrm{Im}z < 0$, such that only solutions with $\sum_{\ell=1}^{K}\sigma_\ell a_\ell > 0$ contribute.

The next task is to perform the integral over $\mathrm{Re}(u)$. It is useful to define the complex variables

$$v_{\ell,\ell+1} = e^{2i\pi U_{\ell,\ell+1}} = e^{i\beta(V_{\ell,\ell+1} - i\Sigma^\star_{\ell,\ell+1})}, \tag{4.31}$$

with fixed modulus $|v_{\ell,\ell+1}| = e^{\beta\Sigma^\star_{\ell,\ell+1}}$. The product of these variables satisfies

$$\prod_{\ell=1}^{K} v_{\ell,\ell+1} = y^2, \tag{4.32}$$

in view of the definition of $U$ in (3.14) and the fact that $R$-charges of chiral fields in an oriented cycle sum up to $R = 2$. In terms of the variables (4.31), for a cyclic quiver with $K$ nodes, (3.34) becomes

$$\mathcal{I}_{\mathrm{sc}} = \left(\frac{-1}{y - 1/y}\right)^{K-1}\sum_{s\in S}\mathrm{sgn}(\det\partial_a\partial_b\widetilde{W})\oint\prod_{\ell=1}^{K-1}\frac{dv_{\ell,\ell+1}}{2\pi i\, v_{\ell,\ell+1}}\left(\prod_{\ell=1}^{K}\left[g_{\mathrm{chiral}}(v_{\ell,\ell+1},0)\right]^{a_\ell}\right), \tag{4.33}$$

where the integral runs over the product of the circles $|v_{\ell,\ell+1}| = e^{\beta\Sigma^\star_{\ell,\ell+1}}$, and $g_{\mathrm{chiral}}$ is the meromorphic function

$$g_{\mathrm{chiral}}(v_{\ell,\ell+1},0) = -y^{-1}\frac{v_{\ell,\ell+1} - y^2}{v_{\ell,\ell+1} - 1}. \tag{4.34}$$

The variable $v_{n,1}$ is understood to be substituted in terms of the remaining $v_{\ell,\ell+1}$'s using (4.32). The sign appearing in (4.33) can be evaluated using (3.22) leading to

$$\mathrm{sgn}(\det\partial_a\partial_b\widetilde{W}) = (-1)^{K-1}\mathrm{sgn}\left(\sum_{i=1}^{K}a_i\sigma_i\right)\prod_{\ell=1}^{K}\sigma_\ell. \tag{4.35}$$

Since the integrand in (4.33) is holomorphic in $v_{\ell,\ell+1}$, the integral may be evaluated by residues, with the modulus of $|v_{\ell,\ell+1}|$ dictating which poles contribute. We carry out this computation for $K = 3$ in Appendix C. Here we adopt a different approach, which easily extends to any $K$.

In order to evaluate the integral over the phase of $v_{\ell,\ell+1}$, we simply expand each of the factors, in the limit $v_{\ell,\ell+1} \to \infty$ whenever $\Sigma^\star_{\ell,\ell+1} > 0$ or $v_{\ell,\ell+1} \to 0$ whenever $\Sigma^\star_{\ell,\ell+1} < 0$. Both cases are covered by the formula

$$g_{\mathrm{chiral}}(v_{\ell,\ell+1},0) = -\sum_{\substack{\alpha=\pm1\\m\geq0}} y^{-\sigma_\ell\alpha}\, v_{\ell,\ell+1}^{-\sigma_\ell\left(m+\frac{1-\alpha}{2}\right)}. \tag{4.36}$$

---

[14] We assume that $\sum_{\ell=1}^{K}\sigma_\ell a_\ell$ never vanishes. Non-generic cases where $\sum_{\ell=1}^{K}\sigma_\ell a_\ell$ vanishes for some choices of signs can be treated by perturbing the $a_\ell$'s. We expect that the index is a continuous function of the $a_\ell$'s such that the result is independent of the choice of perturbation.

Performing this expansion for each factor in (4.33), the integrand becomes

$$(-1)^{\sum_{\ell=1}^{K} a_\ell} \sum_{\substack{\vec{m}_1,\dots,\vec{m}_K \in \mathbb{N} \\ \vec{a}_1,\dots,\vec{a}_K \in \{\pm 1\}}} (-1)^{\sum_{\ell=1}^{K} \frac{a_\ell - A_\ell}{2}} y^{-\sum_{\ell=1}^{K} \sigma_\ell A_\ell - 2\sigma_K \left( M_K + \frac{a_K - A_K}{2} \right)}$$

$$\times \prod_{\ell=1}^{K-1} v_{\ell,\ell+1}^{\sigma_K \left( M_K + \frac{a_K - A_K}{2} \right) - \sigma_\ell \left( M_\ell + \frac{a_\ell - A_\ell}{2} \right)}, \tag{4.37}$$

where the notations are as follows: each $\vec{m}_\ell$ is a $a_\ell$-dimensional vector with entries in non-negative integers, while each $\vec{a}_\ell$ is a $a_\ell$-dimensional vector with entries in $\pm 1$. Moreover, $M_\ell$ and $A_\ell$ are the sums of the components of $\vec{m}_\ell$ and $\vec{a}_\ell$, respectively. The integral over phases picks up terms with vanishing powers of all $v_{\ell,\ell+1}$'s, i.e. such that

$$\sigma_\ell \left( M_\ell + \frac{a_\ell - A_\ell}{2} \right) = \sigma_K \left( M_K + \frac{a_K - A_K}{2} \right) \qquad \forall \ell = 1,\dots,K-1. \tag{4.38}$$

To proceed, we distinguish two different types of contributions, whether all signs $\sigma_i$ are equal or not. Correspondingly,

$$\mathcal{I}_{\text{sc}} = \mathcal{I}_{\text{same}} + \mathcal{I}_{\text{uneq}}, \tag{4.39}$$

where $\mathcal{I}_{\text{same}}$ and $\mathcal{I}_{\text{uneq}}$ are as follows.

- If the signs $\sigma_i$ are not all equal, the expressions inside parentheses in (4.38) are non-negative, and therefore (4.38) is satisfied if and only if the expressions inside parentheses vanish individually. Since $M_\ell$ and $a_\ell - A_\ell$ are non-negative, we see that all the entries of the vectors $\vec{m}_\ell$ and $\vec{a}_\ell$ are 0 and 1, respectively. For these solutions,

$$\mathcal{I}_{\text{uneq}}(\{a_\ell\}, y) = \frac{(-1)^{\sum_{\ell=1}^{K} a_\ell}}{(y - 1/y)^{K-1}} \sideset{}{'}\sum_{\substack{\sigma_\ell = \pm 1 \\ \sum_{\ell=1}^{K} a_\ell \sigma_\ell > 0}} \left( \prod_{\ell=1}^{K} \sigma_\ell \right) y^{-\sum_{\ell=1}^{K} a_\ell \sigma_\ell}, \tag{4.40}$$

where the prime indicates that the term with equal signs $\sigma_\ell$ is excluded.

- If all signs $\sigma_\ell$ are equal, then the constraints (4.38) are less stringent, and simply require that $M_\ell + \frac{a_\ell - A_\ell}{2}$ is the same for all $\ell = 1,\dots K$. We denote by $M$ this common value,

$$M_\ell + \frac{a_\ell - A_\ell}{2} = M \qquad \forall \ell = 1,\dots,K. \tag{4.41}$$

Introducing $p_\ell = \frac{a_\ell - A_\ell}{2}$, the integrand (4.37) becomes

$$(-1)^{\sum_{\ell=1}^{K} a_\ell} \sum_{\substack{\vec{m}_1,\dots,\vec{m}_K \in \mathbb{N} \\ \vec{a}_1,\dots,\vec{a}_K \in \{\pm 1\}}} (-1)^{\sum_{\ell=1}^{K} p_\ell} y^{-\sum_{\ell=1}^{K} (a_\ell - 2p_\ell) - 2M} \prod_{\ell=1}^{K} \delta_{M_\ell, M - p_\ell}$$

$$= (-y)^{-\sum_{\ell=1}^{K} a_\ell} \sum_{\substack{\vec{m}_1,\dots,\vec{m}_K \in \mathbb{N} \\ \vec{a}_1,\dots,\vec{a}_K \in \{\pm 1\}}} y^{-2M} \prod_{\ell=1}^{K} (-y^2)^{p_\ell} \delta_{M_\ell, M - p_\ell}. \tag{4.42}$$

The sum over $\vec{m}_\ell, \vec{a}_\ell$ can be traded for a sum over $M \geq 0$ and $p_\ell \geq 0$, at the cost of introducing a measure factor $\binom{a}{p}\binom{M-p+a-1}{a-1}$, coming from the number of choices of $\vec{a}_\ell$ and $\vec{m}_\ell$, respectively. For $(\sigma_1,\dots,\sigma_K) = (1,\dots,1)$ (the appropriate choice when $\text{Im} z < 0$), we get

$$\mathcal{I}_{\text{same}} = \frac{(-y)^{-\sum_{\ell=1}^{K} a_\ell}}{(y - 1/y)^{K-1}} \left( 1 + \sum_{M=1}^{\infty} y^{-2M} \prod_{\ell=1}^{K} N_{a_\ell}(M; y) \right), \tag{4.43}$$

where

$$N_a(M; y) = \sum_{p=0}^{a} \frac{a(M+a-p-1)!}{p!(a-p)!(M-p)!}(-y^2)^p.$$

(4.44)

Given that the object (4.44) governs the dominant contribution to the index for large $a_i$'s, it is worth commenting on its properties. By relaxing the constraint $p \leq a$ in the sum, it can be expressed as a hypergeometric series and in turn recognized as a Jacobi polynomial,

$$\begin{aligned}
N_a(M, y) &= \frac{a(M+a-1)!}{a!M!} \, {}_2F_1\left(-a, -M; 1-a-M; y^2\right) \\
&= \frac{(-1)^a a}{M} P_a^{(-a-M,-1)}(1-2y^2).
\end{aligned}$$

(4.45)

This expression holds only for $M > 0$, whereas for $M = 0$ one has $N_a(0, y) = 1$ for any $a$. Another useful representation is

$$N_a(M, y) = a(1-y^2)\mathcal{M}_{a-1}(M-1, 2, 1/y^2),$$

(4.46)

where $\mathcal{M}_n(x, \beta, c) = {}_2F_1(-n, -x; \beta; 1-1/c)$ are the Meixner polynomials [38], which are discrete analogues of the generalized Laguerre polynomials $L_m^\alpha(x)$. Thus, the infinite sum in (4.43) can be viewed as the refined counterpart of the integral representation of the unrefined index for cyclic quivers in chamber $\mathcal{C}$,

$$\Omega(\{a_\ell\}, 1) = (-1)^{1+\sum_{\ell=1}^{K} a_\ell}\left[\prod_{\ell=1}^{K-1} a_\ell - \int_0^\infty ds\, e^{-s} \prod_{\ell=1}^{K} L_{a_\ell-1}^1(s)\right].$$

(4.47)

This formula generalizes [41, (E.2)] to the case of cyclic quivers with an arbitrary number of nodes, and can be derived from (4.13) by representing the second term as an integral,

$$Z_{\mathcal{C}}(1) = -\frac{x_K}{1+x_K}\prod_{\ell=1}^{K-1}\frac{x_\ell}{(1+x_\ell)^2} + \int_0^\infty ds\, e^{-s}\prod_{\ell=1}^{K}\frac{x_\ell}{(1+x_\ell)^2} e^{-s-\sum_{\ell=1}^{K}\frac{sx_\ell}{1+x_\ell}}$$

(4.48)

and Taylor expanding the exponential using $\frac{e^{-\frac{sx}{1-x}}}{(1-x)^{\alpha+1}} = \sum_{n=0}^\infty L_n^\alpha(s)x^n$.

## 4.6 Generating series for scaling invariants

While the formulae above are easily evaluated for specific choices of $a_\ell$'s, it will be useful to obtain generating series of the above contributions, similar to §3.4. For this purpose we need the generating series of (4.44). For $M > 0$ this is easily obtained by exchanging the sums,

$$\begin{aligned}
\sum_{a=1}^\infty N_a(M, y)x^a &= \sum_{p=0}^\infty \sum_{m=0}^\infty \frac{(p+m)(M+m-1)!}{p!m!(M-p)!}(-y^2x)^p x^m \\
&= \frac{x(1-y^2)}{(1-x)(1-xy^2)}\left(\frac{1-xy^2}{1-x}\right)^M.
\end{aligned}$$

(4.49)

For $M = 0$ the r.h.s. should be replaced by $x/(1-x)$.

From (4.43) we can compute the generating function

$$
\begin{aligned}
Z_{\text{same}} &= \frac{1}{(y-1/y)^{K-1}} \left[ \prod_{\ell=1}^{K} \frac{(-x_\ell/y)}{1+x_\ell/y} + \sum_{M=1}^{\infty} y^{-2M} \prod_{\ell=1}^{K} \frac{x_\ell(y-1/y)}{(1+x_\ell/y)(1+x_\ell y)} \left( \frac{1+x_\ell y}{1+x_\ell/y} \right)^M \right] \\
&= \frac{1}{(y-1/y)^{K-1}} \left[ \prod_{\ell=1}^{K} \frac{(-x_\ell/y)}{1+x_\ell/y} + \left( \prod_{\ell=1}^{K} \frac{x_\ell(y-1/y)}{(1+x_\ell/y)(1+x_\ell y)} \right) \times \frac{y^{-2} \prod_{\ell=1}^{K} \frac{1+x_\ell y}{1+x_\ell/y}}{1-y^{-2} \prod_{\ell=1}^{K} \frac{1+x_\ell y}{1+x_\ell/y}} \right] \\
&= \frac{1}{(y-1/y)^{K-1}} \left[ (-1/y)^K + \frac{y^{-1}(y-1/y)^K}{y \prod_{\ell=1}^{K}(1+x_\ell/y) - 1/y \prod_{\ell=1}^{K}(1+x_\ell y)} \right] \prod_{\ell=1}^{K} \frac{x_\ell}{1+x_\ell/y} .
\end{aligned}
\tag{4.50}
$$

It is worth remarking that this is precisely the generating function (4.22) of trivial stability indices. We shall return to this observation below.

We now turn to the contribution (4.40) from unequal signs. While it is possible to compute the generating function for low values of $K$ (see appendix §D), it seems hard to construct it for any $K$. Instead, a more efficient strategy is to combine $Z_{\text{uneq}}$ with the generating series $Z_{\text{reg}}$ for regular collinear solutions. The contribution from regular collinear solutions at finite $\zeta$, in the chamber $\mathcal{C}$ where $\zeta_1, \ldots, \zeta_{K-1} > 0, \zeta_K < 0$, is given by [12, (4.13)]:

$$
\mathcal{I}_{\text{reg}}(\{a_\ell\}, y) = \frac{(-1)^{K-1+\sum_{\ell=1}^{K} a_\ell}}{(y-1/y)^{K-1}} \sum_{\substack{\sigma_\ell = \pm 1 \\ \text{sgn}(\sum_{\ell=1}^{K} a_\ell \sigma_\ell) = -\text{sgn}(\sigma_K)}} \left( \prod_{\ell=1}^{K-1} \sigma_\ell \right) y^{\sum_{\ell=1}^{K} a_\ell \sigma_\ell}, \tag{4.51}
$$

where the contribution from equal signs trivially vanishes. Rewriting (4.40) as

$$
\mathcal{I}_{\text{uneq}}(\{a_i\}, y) = -\frac{(-1)^{K-1+\sum_{i=1}^{K} a_i}}{(y-1/y)^{K-1}} \sideset{}{'}\sum_{\substack{\sigma_i = \pm 1 \\ \sum_{i=1}^{K} a_i \sigma_i < 0}} \left( \prod_{i=1}^{K} \sigma_i \right) y^{\sum_{i=1}^{K} a_i \sigma_i} \tag{4.52}
$$

we see that the contributions with $\sigma_K = +1$ cancel in the sum of unequal sign and regular contributions. This leaves only the contribution from $\sigma_K = -1$, and $\sigma_\ell$ not all equal to $-1$, with no condition on the sign of $\sum_{\ell=1}^{K} a_\ell \sigma_\ell$:

$$
\begin{aligned}
\mathcal{I}_{\text{reg}}(\{a_\ell\}, y) + \mathcal{I}_{\text{uneq}}(\{a_\ell\}, y) &= \frac{(-1)^{K-1+\sum a_\ell}}{(y-1/y)^{K-1}} \sum_{\substack{\sigma_\ell = \pm 1 \\ (\sigma_1, \ldots, \sigma_{K-1}) \neq (-1, \ldots, -1)}} \left( \prod_{\ell=1}^{K-1} \sigma_\ell \right) y^{-a_K + \sum_{\ell=1}^{K-1} a_\ell} \\
&= \frac{(-1)^{K-1+\sum a_\ell}}{(y-1/y)^{K-1}} y^{-a_K} \left[ \prod_{\ell=1}^{K-1} (y^{a_\ell} - y^{-a_\ell}) - (-1)^{K-1} y^{-(a_1 + \cdots + a_{K-1})} \right]. \tag{4.53}
\end{aligned}
$$

The generating series of $\mathcal{I}_{\text{reg}} + \mathcal{I}_{\text{uneq}}$ is easily constructed,

$$
Z_{\text{reg}} + Z_{\text{uneq}} = -\frac{1}{y} \prod_{\ell=1}^{K} \frac{x_\ell}{(1+x_\ell/y)(1+x_\ell y)} \left[ (1+x_K y) - \frac{(-1/y)^{K-1}}{(y-1/y)^{K-1}} \prod_{\ell=1}^{K} (1+x_\ell y) \right]. \tag{4.54}
$$

Collecting all contributions and after some algebra, we finally arrive at

$$
\begin{aligned}
Z_{\text{reg}} + Z_{\text{uneq}} + Z_{\text{same}} &= -x_K \prod_{l=1}^{K-1} \frac{x_l}{(1+x_l/y)(1+x_l y)} \frac{\prod_{\ell=1}^{K-1}(1+x_\ell/y) - \prod_{\ell=1}^{K-1}(1+x_\ell y))}{y \prod_{\ell=1}^{K}(1+x_\ell/y) - 1/y \prod_{\ell=1}^{K}(1+x_\ell y)} \\
&= \left[ \frac{y \prod_{\ell=1}^{K-1}(1+x_\ell/y) - y^{-1} \prod_{\ell=1}^{K-1}(1+yx_\ell)}{y \prod_{\ell=1}^{K}(1+x_\ell/y) - y^{-1} \prod_{\ell=1}^{K}(1+yx_\ell)} - 1 \right] \prod_{\ell=1}^{K-1} \frac{x_\ell}{(1+x_\ell y)(1+x_\ell/y)}, \tag{4.55}
\end{aligned}
$$

which precisely matches (4.12). We conclude that that the sum of the deep scaling and regular contributions produces the correct total index, including contributions from single-centered and scaling solutions.

In fact, we claim that the deep scaling region alone produces the single-centered index, up to a contribution from the minimal modification hypothesis,

$$\mathcal{I}_{\text{same}} + \mathcal{I}_{\text{uneq}} = \Omega_{\text{S}} + H \,, \tag{4.56}$$

where $H$ is defined as the minimal modification of the Coulomb index $g_C(\{\gamma_1, \dots \gamma_K\})$, or equivalently as the minimal modification of the regular part $\mathcal{I}_{\text{reg}}$. Since $\mathcal{I}_{\text{same}}$ coincides with the stacky invariant for trivial stability $\mathcal{A}_0$ (after dividing by $y - 1/y$ and changing $y \to -1/w$), this is equivalent upon using (B.14) to

$$Z_{\text{uneq}} - Z_H \;\; = \;\; \prod_{\ell=1}^{K} \frac{x_\ell}{(1 + x_\ell/y)} \left[ -\frac{y^{-K}}{(y - 1/y)^{K-1}} + \frac{P_K(\{x_\ell\}; -1/y)}{\prod_{1 \le k < \ell \le K}(1 - x_k x_\ell)} \right]. \tag{4.57}$$

In §D we verify this identity explicitly for $K = 3$ and $K = 4$.

We conclude with a remark for the mathematically minded reader. Since $H$ is in the kernel of the projection operator (2.20) and since $\Omega_{\text{S}}$ is a symmetric Laurent polynomial, hence unaffected by this projection, it follows from Eq. (4.56) that

$$\Omega_{\text{S}} = M\left[\mathcal{I}_{\text{same}}\right] + M\left[\mathcal{I}_{\text{uneq}}\right]. \tag{4.58}$$

Now, recall that our observation below (4.50) that $\mathcal{I}_{\text{same}}$ is equal (up to redefinition $y \to 1/w$ and a factor $(y - 1/y)$) to the stacky invariant with trivial stability $\mathcal{A}_0$, which is a well-defined mathematically. Hence, in order to put $\Omega_{\text{S}}$ on solid mathematical footing, it would suffice to establish the mathematical meaning of the still mysterious part $\mathcal{I}_{\text{uneq}}$.

## Acknowledgments

We are grateful to Pierre Descombes, Philippe Di Francesco, Mourad Ismaïl, Jan Manschot, Sergey Mozgovoy, Ashoke Sen, Piljin Yi and Jean-Bernard Zuber for useful discussions related to parts of this project. The research of SM is supported by Laureate Award 15175 "Modularity in Quantum Field Theory and Gravity" of the Irish Research Council.

## A  Geometric condition on existence of scaling solutions

In this section, we show that the condition (2.12) for existence of scaling solutions holds in the case of a cyclic quiver with one additional arrow, say $\kappa_{1,k} > 0$ with $k \neq 2$ and $k \neq K$. In that case, it is straightforward to show that scaling solutions exist if and only if

$$
\begin{aligned}
&\forall j < k \,, \sum_{\substack{i=1\dots K \\ i \neq j}} \kappa_{i,i+1} > \kappa_{j,j+1} + \kappa_{1,k} \,, \\
&\forall j \ge k \,, \sum_{\substack{i=1\dots K \\ i \neq j}} \kappa_{i,i+1} > \kappa_{j,j+1} - \kappa_{1,k} \,.
\end{aligned}
\tag{A.1}
$$

To establish this, we note that the equations (2.10) imply that the ratios $\lambda_i = \frac{\kappa_{i\,i+1}}{r_{i\,i+1}}$ with $r_{ij} : |\vec{x}_i - \vec{x}_j|$ can take only two values, namely $\lambda_i = \lambda_1$ for $1 \le i < k$ and $\lambda_i = \lambda_k$ for

$k \leq i \leq K$. Moreover $\lambda_k - \lambda_1 = \kappa_{1k}/r_{1k} > 0$. The existence of the $k$-sided polygon with vertices $\vec{x}_1, \vec{x}_2, \ldots, \vec{x}_k$ requires

$$\sum_{i=1}^{k-1} r_{i\,i+1} \geq r_{1k} \quad \text{and} \quad \forall j < k, \sum_{i=1}^{k-1} r_{i\,i+1} + r_{1k} \geq 2r_{j\,j+1}, \tag{A.2}$$

and similarly the existence of the $(n+2-k)$-sided polygon going through the points $\vec{x}_1, \vec{x}_k, \vec{x}_{k+1}$, ..., $\vec{x}_K$ requires

$$\sum_{i=k}^{K} r_{i\,i+1} \geq r_{1k} \quad \text{and} \quad \forall j \geq k, \sum_{i=k}^{K} r_{i\,i+1} + r_{1k} \geq 2r_{j\,j+1}. \tag{A.3}$$

Expressing the distances in terms of $\lambda_1, \lambda_k$, these constraints become

$$\begin{aligned}
\forall j < k \quad & \sum_{i=1}^{k-1} \kappa_{i\,i+1} + \kappa_{1k} \geq a\kappa_{1k} \geq 2\kappa_{j\,j+1} - \sum_{i=1}^{k-1} \kappa_{i\,i+1} + \kappa_{1k}, \\
\forall j \geq k \quad & \sum_{i=k}^{K} \kappa_{i\,i+1} \geq a\kappa_{1k} \geq 2\kappa_{j\,j+1} - \sum_{i=k}^{K} \kappa_{i\,i+1},
\end{aligned} \tag{A.4}$$

where $a = \frac{\lambda_k}{\lambda_k - \lambda_1} > 1$. The existence of a number $a\kappa_{1k}$ satisfying both inequalities implies the two conditions in (A.1). QED.

# B Computing indices for cyclic quivers

## B.1 Trivial stability invariants

The Reineke formula expresses the stacky invariants $\mathcal{A}(\gamma; w)$ for stability $\zeta$ (defined in (4.16)) in terms of the stacky invariants $\mathcal{A}_0(\gamma; w)$ for trivial stability condition as follows [27]:

$$\mathcal{A}(\gamma; w) = \sum_{\substack{\alpha_1 + \cdots + \alpha_k = \gamma, k \geq 1 \\ \mu(\sum_{j=1}^{m} \alpha_j) > \mu(\gamma), m=1,\ldots,k-1}} (-1)^{k-1} w^{-\sum_{i<j} \langle \alpha_i, \alpha_j \rangle} \prod_{j=1}^{k} \mathcal{A}_0(\alpha_j, w). \tag{B.1}$$

Conversely, if we know the invariants $\mathcal{A}(\gamma; w)$ for a given stability condition $\zeta$, we can use it to compute the stacky invariants for trivial stability.

To perform this computation for a cyclic quiver with dimension vector $\gamma = (1, 1, \ldots, 1)$, we use the following observation from [12]: for vanishing superpotential, the stacky invariants in the chamber $\mathcal{C}$ are given by

$$\mathcal{A}_0(\gamma_1 + \cdots + \gamma_K) = \frac{w^{a_K} \prod_{\ell=1}^{K-1} (w^{a_\ell} - w^{-a_\ell})}{(w - 1/w)^K}, \tag{B.2}$$

corresponding to the fact that the quiver moduli space is a product of projective and affine spaces $\prod_{\ell=1}^{K-1} \mathbb{P}^{a_\ell-1} \times \mathbb{C}^{a_K}$. This result follows by applying (B.1) with $h(\alpha_j, w)$ substituted by $\mathcal{A}_0^{W=0}(\alpha_i; w)$ given by (4.19),

$$\mathcal{A}_0^{W=0}(\gamma_1 + \cdots + \gamma_K) = \frac{w^{\sum_{\ell=1}^{K} a_\ell}}{(w - 1/w)^K}. \tag{B.3}$$

Since $\mathcal{A}_0^{W=0}(\alpha, j, w) = \mathcal{A}_0(\alpha_j, w)$ for vectors which are not supported on all nodes (as the superpotential constraint become trivial in such cases), it follows that

$$\mathcal{A}(\gamma_1 + \cdots + \gamma_K) - \mathcal{A}_0(\gamma_1 + \cdots + \gamma_K) = \mathcal{A}^{W=0}(\gamma_1 + \cdots + \gamma_K) - \mathcal{A}_0^{W=0}(\gamma_1 + \cdots + \gamma_K). \quad \text{(B.4)}$$

Since we already know the generating series of $\mathcal{A}$ from (4.12), it suffices to compute the generating series of $\Delta\mathcal{A} := \mathcal{A}^{W=0} - \mathcal{A}_0^{W=0}$. The latter is given by

$$
\begin{aligned}
Z_{\Delta\mathcal{A}}(x_i, w) &= \frac{\prod_{\ell=1}^{K} wx_\ell}{(w-1/w)^K \prod_{\ell=1}^{K}(1-wx_\ell)} - \frac{w\prod_{\ell=1}^{K} x_\ell}{(1-wx_K)(w-1/w)\prod_{\ell=1}^{K-1}(1-wx_\ell)(1-x_\ell/w)} \\
&= \frac{w\prod_{\ell=1}^{K} x_\ell}{(w-1/w)\prod_{\ell=1}^{K}(1-wx_\ell)} \left[ \frac{w^{K-1}}{(w-1/w)^{K-1}} - \frac{1}{\prod_{\ell=1}^{K-1}(1-x_\ell/w)} \right]. \quad \text{(B.5)}
\end{aligned}
$$

Now, from (4.12) it follows that

$$
\begin{aligned}
Z_{\mathcal{A}}(x_i, w) &= \frac{\prod_{\ell=1}^{K-1} x_\ell}{(w-1/w)\prod_{\ell=1}^{K-1}(1-x_\ell w)(1-x_\ell/w)} \\
&\quad \times \left[ \frac{1/w\prod_{\ell=1}^{K-1}(1-x_\ell w) - w\prod_{\ell=1}^{K-1}(1-x_\ell/w)}{1/w\prod_{\ell=1}^{K}(1-x_\ell w) - w\prod_{\ell=1}^{K}(1-x_\ell/w)} - 1 \right]. \quad \text{(B.6)}
\end{aligned}
$$

We therefore deduce the stacky invariants for generic superpotential and trivial stability,

$$
\begin{aligned}
Z_{\mathcal{A}_0}(x_i, w) &= Z_{\mathcal{A}}(x_i, w) + Z_{\Delta\mathcal{A}}(x_i, w) \quad &\text{(B.7)} \\
&= \frac{\prod_{\ell=1}^{K} x_\ell}{\prod_{\ell=1}^{K}(1-wx_\ell)} \left[ \frac{w^K}{(w-1/w)^K} + \frac{w}{1/w\prod_{\ell=1}^{K}(1-wx_\ell) - w\prod_{\ell=1}^{K}(1-x_\ell/w)} \right],
\end{aligned}
$$

which is manifestly symmetric under permutations. Using the identities

$$\prod_{\ell=1}^{K}(1-wx_\ell) = \sum_{\ell=0}^{K}(-w)^\ell \tau_\ell, \quad \text{(B.8)}$$

$$\frac{w\prod_{\ell=1}^{K}(1-x_\ell/w) - 1/w\prod_{\ell=1}^{K}(1-wx_\ell)}{w-1/w} = 1 + \sum_{\ell=1}^{K-1} \frac{(-w)^\ell - (-w)^{-\ell}}{w-1/w}\tau_{\ell+1}, \quad \text{(B.9)}$$

where $\tau_\ell$ are the symmetric functions of $x_i$ (with $\tau_0 = 1$), we finally obtain

$$Z_{\mathcal{A}_0}(x_i, w) = \prod_{\ell=1}^{K} \frac{x_\ell}{(1-wx_\ell)} \left[ \left(\frac{w}{w-1/w}\right)^K - \frac{w}{(w-1/w)} \frac{1}{1 + \sum_{\ell=1}^{K-1} \frac{(-w)^\ell - (-w)^{-\ell}}{w-1/w}\tau_{\ell+1}} \right]. \quad \text{(B.10)}$$

Since the first term is $Z_{\mathcal{A}_0^{W=0}}$, this establishes (4.20).

## B.2 Comparison to single-centered indices

Let us now compare $Z_{\mathcal{A}_0}$ with the generating series of single-centered indices given in (4.2). Setting $Z_{\mathcal{A}_S}(x_i, w) = Z_S(x_i, -1/w)/(w-1/w)$ we find

$$
\begin{aligned}
Z_{\mathcal{A}_S} - Z_{\mathcal{A}_0} &= -\left(\frac{w}{w-1/w}\right)^K \prod_{\ell=1}^{K} \frac{x_\ell}{(1-wx_\ell)} + \frac{1}{2}\prod_{\ell=1}^{K} \frac{x_\ell}{(1-wx_\ell)(1-x_\ell/w)} \quad &\text{(B.11)} \\
&\quad + \frac{1}{2(w-1/w)}\sum_{k=1}^{K} \frac{1-x_k^2}{(1-wx_k)(1-x_k/w)} \prod_{\substack{\ell=1\ldots K \\ \ell \neq k}} \frac{x_\ell}{(1-x_\ell/x_k)(1-x_\ell x_k)}.
\end{aligned}
$$

One can check that the poles at $w = 1/x_\ell$ cancel between the second and third terms, and so do the poles at $x_k = x_\ell$ in the third term. Indeed, the third term can be viewed as the contribution of the poles at $u = x_k$ in the contour integral $\oint_C h(u)\mathrm{d}u$ with

$$h(u) = \frac{(u-1/u)^2}{2(w-1/w)(1-uw)(1-u/w)} \prod_{\ell=1}^{K} \frac{x_\ell}{(1-ux_\ell)(1-x_\ell/u)} . \tag{B.12}$$

This function satisfies $h(1/u) = u^2 h(u)$, is regular at $u = 0$ and $u = \infty$ for $K \geq 2$ and has simple poles at $u = x_\ell$, $u = 1/x_\ell$, $u = w$, $u = 1/w$, with opposite residues at $x$ and $1/x$, or $w$ and $1/w$. Singularities as $x_\ell$ and $w$ vary can only arise when the contour is pinched. Since the contour $\mathcal{C}$ surrounds all $x_\ell$'s, there can be non singularities at $x_k = x_\ell$. Moreover, the second term in (B.11) arises by extending the contour such that it includes the pole at $u = w$, so the sum of the second and third terms must be regular as $x_\ell = w$. Let us define

$$\begin{aligned}
P_K(\{x_\ell\}, w) &= \frac{1}{2}(w-1/w)\frac{\prod_{1\leq k<\ell\leq K}(1-x_k x_\ell)}{\prod_{\ell=1}^{K}(1-x_\ell/w)} \\
&\quad + \frac{1}{2}\sum_{k=1}^{K} \frac{1/x_k - x_k}{1 - x_k/w} \prod_{\substack{\ell=1\ldots K \\ \ell\neq k}} \frac{1-x_\ell w}{1-x_\ell/x_k} \prod_{\substack{1\leq\ell<m\leq K \\ \ell\neq k, m\neq k}} (1-x_\ell x_m),
\end{aligned} \tag{B.13}$$

so that

$$Z_{\mathcal{A}_S} - Z_{\mathcal{A}_0} = \prod_{\ell=1}^{K} \frac{x_\ell}{(1-wx_\ell)}\left[ -\left(\frac{w}{w-1/w}\right)^K + \frac{P_K(\{x_\ell\}, w)}{(w-1/w)\prod_{1\leq k<\ell\leq K}(1-x_k x_\ell)} \right]. \tag{B.14}$$

We shall now prove that $P_K$ satisfies the same recursion and initial value as (4.5). and is therefore the polynomial introduced in that equation.

To show this, let us define

$$\begin{aligned}
A_K &= \frac{1}{2}\prod_{\ell=1}^{K} \frac{x_\ell}{(1-wx_\ell)(1-x_\ell/w)} \tag{B.15} \\
&\quad + \frac{1}{2(w-1/w)}\sum_{k=1}^{K} \frac{1-x_k^2}{(1-wx_k)(1-x_k/w)} \prod_{\substack{\ell=1\ldots K \\ \ell\neq k}} \frac{x_\ell}{(1-x_\ell/x_k)(1-x_\ell x_k)} .
\end{aligned}$$

It is straightforward to show that

$$\begin{aligned}
A_{K+1} &= \frac{x_{K+1}A_K}{(1-wx_{K+1})(1-x_{K+1}/w)} \tag{B.16} \\
&\quad + \frac{1}{2(w-1/w)}\sum_{k=1\ldots K+1} \frac{1-x_k^2}{(1-wx_{K+1})(1-x_{K+1}/w)} \frac{x_{K+1}}{x_k} \prod_{\substack{\ell=1\ldots K+1 \\ \ell\neq k}} \frac{x_\ell}{(1-x_\ell/x_k)(1-x_\ell x_k)} .
\end{aligned}$$

Expressing $P_K$ in terms of $A_K$, this implies that

$$\begin{aligned}
P_{K+1} &= \frac{\prod_{\ell=1}^{K}(1-x_\ell x_{K+1})}{1-x_{K+1}/w}P_K \\
&\quad + \frac{x_{K+1}\prod_{\ell=1}^{K}(1-wx_\ell)}{2(1-x_{K+1}/w)}\sum_{k=1\ldots K+1} (1-1/x_k^2) \frac{\prod_{\substack{1\leq\ell<m\leq K+1 \\ \ell\neq k, m\neq k}}(1-x_\ell x_m)}{\prod_{\substack{\ell=1\ldots K+1 \\ \ell\neq k}}(1-x_\ell/x_k)} . \tag{B.17}
\end{aligned}$$

Let us define the symmetric function

$$S_K(\{x_\ell\}) \;=\; \frac{1}{2} \sum_{k=1...K} \left(1 - 1/x_k^2\right) \frac{\prod_{\substack{1 \le \ell < m \le K \\ \ell \neq k, m \neq k}} (1 - x_\ell x_m)}{\prod_{\substack{\ell = 1...K \\ \ell \neq k}} (1 - x_\ell/x_k)}, \tag{B.18}$$

such that

$$P_{K+1} = \frac{P_K \prod_{\ell=1}^K (1 - x_\ell x_{K+1}) - x_{K+1} S_{K+1} \prod_{\ell=1}^K (1 - w x_\ell)}{1 - x_{K+1}/w}. \tag{B.19}$$

Remarkably, the two terms coming from $(1 - 1/x_k^2)$ in (B.18) produce the same result,

$$S_K(\{x_\ell\}) = \sum_{k=1...K} \frac{\prod_{\substack{1 \le \ell < m \le K \\ \ell \neq k, m \neq k}} (1 - x_\ell x_m)}{\prod_{\substack{\ell = 1...K \\ \ell \neq k}} (1 - x_\ell/x_k)}. \tag{B.20}$$

Moreover, one can show that $S_K$ is regular at $x_\ell = x_k$, hence is symmetric polynomial in variables $x_1, \ldots, x_K$. For $K \le 8$, we find

$$\begin{aligned}
S_K \;=\;\; & 1 - \tau_4 + \tau_5(\tau_1 - \tau_5) + \tau_6\Big(1 - \tau_1^2 + \tau_2 + \tau_4 + \tau_1\tau_5 + \tau_6(-1 - \tau_2 + \tau_6)\Big) \\
& + \tau_7\big(\tau_1 + \tau_1^3 - 2\tau_1\tau_2 + \tau_3 - \tau_1\tau_4 - \tau_1^2\tau_5 + 2\tau_2\tau_5 + 2\tau_1\tau_6 + \tau_1\tau_2\tau_6 - \tau_3\tau_6 \\
& - 2\tau_5\tau_6 - \tau_1\tau_6^2\big) + \tau_7^2\big(-2 - \tau_1^2 - \tau_2^2 + \tau_1\tau_3 + \tau_4 + \tau_1\tau_5 + \tau_6 + \tau_2\tau_6\big) \\
& + \tau_7^3\big(-\tau_1 - \tau_3 + \tau_7\big)\Big) + \mathcal{O}(\tau_8). \tag{B.21}
\end{aligned}$$

Furthermore, comparing to (B.13), we see that

$$P_K(\{x_1, \ldots, x_K\}, w) = w S_{K+1}(\{x_1, \ldots, x_K, w\}). \tag{B.22}$$

Thus, (B.19) is in fact a recursion for $P_K$, identical to (4.5). Using (B.21) and (B.22), we see that the initial data for $P_2$ coincide, and therefore the object $P_K$ defined in (B.13) is also the one introduced in (4.3).

## B.3  Generating series of attractor indices

The same idea used to compute the stacky invariants for trivial stability can also be applied to construct the generating series of attractor indices. For vanishing superpotential, the stacky invariants in the attractor chamber coincide with those in the chamber $\mathcal{C}$ given by (B.2), provided $a_K$ is the largest of all $a_\ell$'s. More generally, they are given by

$$\mathcal{A}_\star^{W=0}(\gamma_1 + \cdots + \gamma_K) = \frac{w^{\max(a_\ell)} \prod_{\ell=1}^K (w^{a_\ell} - w^{-a_\ell})}{(w - 1/w)^K (w^{\max(a_\ell)} - w^{-\max(a_\ell)})}. \tag{B.23}$$

We can then compute the generating series $Z_{\mathcal{A}_\star^{W=0}}$, and obtain the generating series $Z_{\mathcal{A}_\star}$ for generic superpotential from the identity $\mathcal{A}_\star - \mathcal{A}_0 = \mathcal{A}_\star^{W=0} - \mathcal{A}_0^{W=0}$, similar to (B.4). For $K = 3$, we find, in absence of superpotential,

$$Z_{\mathcal{A}_\star^{W=0}} \;=\; \frac{w\tau_3\left(1 - \tau_1\tau_3 + 2\tau_3^2 - 2\tau_3 w + \tau_2\tau_3 w - \tau_3^3 w\right)}{(w - 1/w)(1 - w\tau_3)(1 - \tau_3/w)\prod_i (1 - w x_i)\prod_{i<j}(1 - x_i x_j)} \tag{B.24}$$

and therefore, for generic superpotential,

$$Z_{\mathcal{A}_\star} \;=\; \frac{\tau_3^2\left(2 - \tau_2 + \tau_3^2\right)}{(w - 1/w)(1 - w\tau_3)(1 - \tau_3/w)(1 - \tau_2 + (w + 1/w)\tau_3)\prod_{i<j}(1 - x_i x_j)}. \tag{B.25}$$

We can check that $Z_{\mathcal{A}_\star} - Z_{\mathcal{A}_0}$ has no derangement factor in the denominator as expected, but the numerator is unilluminating. The difference $Z_{\mathcal{A}^\star} - Z_{\mathcal{A}_S}$ has also no derangement factor in the denominator but a much simpler numerator, see (4.15).

For $K = 4$, one finds

$$
Z_{\mathcal{A}_\star} = \frac{\tau_4 F_4(x_i, w)}{(w - 1/w) \prod_{1 \le l < m \le 4} (1 - x_l x_m) \prod_{1 \le l < m < n \le 4} (1 - w x_l x_m x_n)(1 - x_l x_m x_n/w)}
$$
$$
\times \frac{1}{(1 - \tau_4)(1 - \tau_4 w^2)(1 - \tau_4/w^2)(1 - \tau_2 + (w + 1/w)\tau_3 - (w^2 + 1 + 1/w^2)\tau_4)},
$$
(B.26)

where $F_4$ is a complicated symmetric polynomial in $x_i$'s. The result for the difference is somewhat simpler, but still complicated:

$$
Z_{\mathcal{A}_\star} - Z_{\mathcal{A}_S} = \frac{-\tau_4 G_4(x_i, w)}{(w - 1/w) \prod_{1 \le l < m \le 4} (1 - x_l x_m) \prod_{1 \le l < m < n \le 4} (1 - w x_l x_m x_n)(1 - x_l x_m x_n/w)}
$$
$$
\times \frac{1}{(1 - \tau_4)(1 - \tau_4 w^2)(1 - \tau_4/w^2)},
$$
(B.27)

with

$$
\begin{aligned}
G_4(x_i, w) = \\
\tau_3 \tau_4 + \tau_4^2 \Big( 1/w + w - 2\tau_1 - 2(w + 1/w)\tau_2 - \tau_3(1/w^2 + w^2) + \tau_3(1/w + w)\tau_1 \\
+ \tau_3 \tau_2 - (1/w + w)\tau_3^2 - \tau_1 \tau_3^2 + \tau_3^3 \Big) \\
+ \tau_4^3 \Big( 2(1/w + w) + (1 + 3/w^2 + 3w^2)\tau_1 + (w + 1/w)^3 \tau_2 + 2\tau_1 \tau_2 - (w + 1/w)^2 \tau_3 \\
- 2\tau_2 \tau_3 - \tau_1 \tau_3^2 + \tau_3^3 \Big) \\
- \tau_4^4 \Big( 4/w^3 + 7/w + 7w + 4w^3 + (w + 1/w)^4 \tau_1 + (w + 1/w)^3 \tau_1^2 + (1 + 2/w^2 + 2w^2)\tau_1 \tau_2 \\
+ (w + 1/w)\tau_2^2 - (7 + 5/w^2 + 5w^2)\tau_3 - (1/w^3 + 5/w + 5w + w^3)\tau_1 \tau_3 \\
- (2 + 1/w^2 + w^2)\tau_1^2 \tau_3 - (-2 + 1/w^2 + w^2)\tau_2 \tau_3 - (1/w + w)\tau_1 \tau_2 \tau_3 \\
+ 2(w + 1/w)\tau_3^2 + (w + 1/w)^2 \tau_1 \tau_3^2 + (w + 1/w)\tau_2 \tau_3^2 \Big) \\
+ \tau_4^5 \Big( 1/w^5 + 5/w^3 + 8/w + 8w + 5w^3 + w^5 + (10 + 3/w^4 + 5/w^2 + 5w^2 + 3w^4)\tau_1 \\
+ (2/w^3 + 3/w + 3w + 2w^3)\tau_1^2 - \tau_2(1/w^3 + 1/w + w + w^3) + (2 + 1/w^2 + w^2)\tau_1 \tau_2 \\
+ (w + 1/w)\tau_1^2 \tau_2 + 2(w + 1/w)\tau_2^2 - (2/w^4 + 7/w^2 + 9 + 7w^2 + 2w^4)\tau_3 \\
- (4/w^3 + 6/w + 6w + 4w^3)\tau_1 \tau_3 - (w + 1/w)^2 \tau_1^2 \tau_3 + (2 + 1/w^2 + w^2)\tau_2 \tau_3 \\
+ (1/w + w)\tau_1 \tau_2 \tau_3 + (2/w^3 + 3/w + 3w + 2w^3)\tau_3^2 + (2 + 1/w^2 + w^2)\tau_1 \tau_3^2 \Big) \\
- \tau_4^6 \Big( 2(1/w^5 + 1/w^3 + 4/w + 4w + w^3 + w^5) + (2/w^4 + 7/w^2 + 9 + 7w^2 + 2w^4)\tau_1 \\
+ 2(w + 1/w)\tau_1^2 - \tau_1^3 + (1/w^3 + 1/w + w + w^3)\tau_2 - (w^2 - 2 + 1/w^2)\tau_1 \tau_2 \\
+ (w + 1/w)\tau_2^2 - (10 + 3/w^4 + 5/w^2 + 5w^2 + 3w^4)\tau_3 - (1/w^3 + 5/w + 5w + w^3)\tau_1 \tau_3 \\
+ \tau_1^2 \tau_3 + (1 + 2/w^2 + 2w^2)\tau_2 \tau_3 + (w + 1/w)^3 \tau_3^2 \Big)
\end{aligned}
$$

$$
\begin{aligned}
&+\tau_4^7\Big(1/w^5 + 5/w^3 + 8/w + 8w + 5w^3 + w^5 + (7 + 5/w^2 + 5w^2)\tau_1 + \tau_1^3 - 2\tau_1\tau_2 \\
&-(w+1/w)^4\tau_3 - \tau_1^2\tau_3 - 2\tau_2\tau_3\Big) \\
&-\tau_4^8\Big(4/w^3 + 7/w + 7w + 4w^3 + (w+1/w)^2\tau_1 + (w+1/w)\tau_1^2 \\
&-(w+1/w)^3\tau_2 - \tau_1\tau_2 - (1 + 3/w^2 + 3w^2)\tau_3 - (1/w + w)\tau_1\tau_3\Big) \\
&+\tau_4^9\Big(2(w+1/w) - (w^2 + 1/w^2)\tau_1 - 2(w+1/w)\tau_2 - 2\tau_3\Big) + \Big(1/w + w + \tau_1\Big)\tau_4^{10}.
\end{aligned}
\tag{B.28}
$$

The Taylor expansion of both $Z_{\mathcal{A}_\star}$ and $Z_{\mathcal{A}_S}$ starts at order $x_i^7$, with the first nontrivial terms corresponding to the following values of $\{a_\ell\}$ (up to permutations),

$$
\begin{aligned}
\Omega_\star(\{1,2,2,2\}) &= 2, \quad \Omega_S(\{1,2,2,2\}) = 1, \\
\Omega_\star(\{2,2,2,2\}) &= 0, \quad \Omega_S(\{2,2,2,2\}) = y + 1/y, \\
\Omega_\star(\{1,3,3,3\}) &= 0, \quad \Omega_S(\{1,3,3,3\}) = y + 1/y.
\end{aligned}
\tag{B.29}
$$

## C Residue computation for three-node abelian cyclic quiver

We consider the 3-node cyclic quiver with $\kappa_{12} = a_1, \kappa_{23} = a_2, \kappa_{31} = a_3$. As explained in (3.34), the contribution of collinear scaling solutions is given by the following residue:

$$
\begin{aligned}
\mathcal{I}_{\mathrm{sc}} = \sum_{\substack{\sigma_1,\sigma_2,\sigma_3 = \pm 1 \\ \sigma_1 a_1 + \sigma_2 a_2 + \sigma_3 a_3 < 0}} \frac{\mathrm{sgn}(\sigma_1 a_1 + \sigma_2 a_2 + \sigma_3 a_3)}{(y - 1/y)^2} \oint \frac{dv_1}{v_1} \oint \frac{dv_2}{v_2} \\
\times \left(\frac{y - v_1/y}{v_1 - 1}\right)^{a_1} \left(\frac{y - v_2/y}{v_2 - 1}\right)^{a_2} \left(\frac{v_1 v_2 - 1}{y - v_1 v_2/y}\right)^{a_3},
\end{aligned}
\tag{C.1}
$$

where the sum runs over the possible signs $\sigma_\ell = \mathrm{sgn}\Sigma^\star_{\ell,\ell+1}$ with $\ell \in \{1,2,3\}$. Defining $v_3 = y^2/(v_1 v_2)$ so as to expose the symmetry of the integrand, the integral runs over the two-torus spanned by the phases of $v_\ell$ subject to the constraint $v_1 v_2 v_3 = y^2$, while the moduli are fixed to $|v_\ell| = e^{2\pi a_\ell \sigma_\ell \lambda \mathrm{Im} z}$ with fixed $\lambda > 0$. In contrast to the body of the paper, here we shall assume that $\mathrm{Im} z > 0$; the result for $\mathrm{Im} z < 0$ can be obtained by flipping the signs $\sigma_\ell$.

We shall first perform the integral over $v_2$. There are 4 poles at $v_2 \in \{0, 1, \frac{y^2}{v_1}, \infty\}$. The pole at 0 is always included inside the contour while the pole at $\infty$ never is. Whether or not the other two are inside the contour depends on the modulus of $v_2$:

- if $|v_2| > 1$ (which occurs when $\sigma_2 = +1$) the pole at $v_2 = 1$ is included.

- if $|v_2| > |y^2/v_1|$ (which occurs when $\sigma_3 = -1$) the pole at $v_2 = y^2/v_1$ is included.

Next we integrate over $v_1$. If the first residue over $v_2$ was taken at 0 or $\infty$, the result has only 3 poles at $v_1 \in \{0, 1, \infty\}$. If instead the first residue was taken at $v_2 = 1$ or $y^2/v_1$, then there is an extra pole at $v_1 = y^2$. While the pole at $v_1 = 0$ is always included inside the contour and the one at $v_1 = \infty$ never is, the remaining ones depend on the modulus of $v_1$:

- If $|v_1| > 1$ (which occurs when $\sigma_1 = +1$) the pole at $v_1 = 1$ is included,

- if $|v_1| > |y^2|$ (which occurs when $\sigma_2 a_2 + \sigma_3 a_3 < 0$) the pole at $v_1 = y^2$ is included.

Moreover, out of the 8 possible choices of sign $(\sigma_1, \sigma_2, \sigma_3)$, only 4 contribute, depending whether the triangular inequalities are obeyed or not. We introduce the notation $R_{v_2,v_1}$ for the corresponding residue.

In the triangular inequalities are violated, say $a_1 > a_2 + a_3$, the following sign choices contribute (omitting an overall factor of $(y - 1/y)^2$) The contribution from equal signs always contributes, irrespective of the triangular inequalities:

$$
\begin{aligned}
(-,-,-) &: \quad R_{0,0} + R_{y^2/v_1,0} + R_{y^2/v_1,y^2}, \\
(-,+,+) &: \quad R_{0,0} + R_{1,0}, \\
(-,+,-) &: \quad -R_{0,0} - R_{1,0} - R_{y^2/v_1,0}, \\
(-,-,+) &: \quad -R_{0,0}.
\end{aligned}
\tag{C.2}
$$

Note that in the third line, there are two additional residues $R_{1,y^2} + R_{y^2/v_1,y^2}$ contributing when $a_2 < a_3$, but their sum vanishes. Summing up these four contributions, only one residue remains:

$$
\begin{aligned}
\mathcal{I}_{\text{sc}} &= R_{y^2/v_1,y^2}/(y - 1/y)^2 \\
&= \frac{1}{(y-1/y)^2} \text{Res}_{v_1 = y^2} \text{Res}_{v_2 = y^2/v_1} \frac{1}{v_1 v_2} \left( \frac{y - v_1/y}{v_1 - 1} \right)^{a_1} \left( \frac{y - v_2/y}{v_2 - 1} \right)^{a_2} \left( \frac{v_1 v_2 - 1}{y - v_1 v_2/y} \right)^{a_3} \\
&= \frac{1}{(y-1/y)^2} \frac{1}{(a_3 - 1)!} \text{Res}_{v_1 = y^2} \left( \frac{d}{dv_2} \right)^{a_3 - 1} \left[ \frac{1}{v_1 v_2} \left( \frac{y - v_1/y}{v_1 - 1} \right)^{a_1} \left( \frac{y - v_2/y}{v_2 - 1} \right)^{a_2} \left( \frac{v_1 v_2 - 1}{-v_1/y} \right)^{a_3} \right] \Big|_{v_2 = y^2/v_1}.
\end{aligned}
\tag{C.3}
$$

In this expression the possible pole at $v_1 = y^2$ comes from $\frac{(y - v_1/y)^{a_1}}{(v_2 - 1)^{a_2}}$. The maximum order possible for this pole arises when all derivatives act on $\frac{(y - v_1/y)^{a_1}}{(v_2 - 1)^{a_2}}$ so we obtain

$$
\frac{(y - v_1/y)^{a_1}}{(v_2 - 1)^{a_2 + a_3 - 1}} \Big|_{v_2 = y^2/v_1} = \frac{(v_1/y)^{a_1} (y^2/v_1 - 1)^{a_1}}{(y^2/v_1 - 1)^{a_2 + a_3 - 1}} = (v_1/y)^{a_1} (y^2/v_1 - 1)^{a_1 - a_2 - a_3 + 1}.
$$

Since $a_1 > a_2 + a_3$, there is no pole and the residue vanishes. Therefore, the scaling index (C.1) vanishes when triangular inequality are violated.

Let us now turn to the case where the triangular inequality are obeyed, $a_1 < a_2 + a_3$, $a_2 < a_1 + a_3$, $a_3 < a_1 + a_2$. In that case, the following sign choices contribute:

$$
\begin{aligned}
(-,-,-) &: \quad R_{0,0} + R_{y^2/v_1,0} + R_{y^2/v_1,y^2}, \\
(+,-,-) &: \quad -R_{0,0} - R_{0,1} - R_{y^2/v_1,0} - R_{y^2/v_1,y^2} - R_{y^2/v_1,1}, \\
(-,+,-) &: \quad -R_{0,0} - R_{1,0} - R_{y^2/v_1,0}, \\
(-,-,+) &: \quad -R_{0,0}.
\end{aligned}
\tag{C.4}
$$

As before, in the third line, there are two additional residues $R_{1,y^2} + R_{y^2/v_1,y^2}$ contributing when $a_2 < a_3$, but their sum vanishes. Summing up these contributions, we get

$$
(y - 1/y)^2 \mathcal{I}_{\text{sc}} = -2R_{0,0} - R_{1,0} - R_{0,1} - R_{y^2/v_1,0} - R_{y^2/v_1,1}.
\tag{C.5}
$$

By deforming the contours adequately, we see that

$$
\begin{aligned}
-R_{0,0} - R_{1,0} - R_{y^2/v_1,0} &= R_{\infty,0} \\
-R_{0,1} - R_{y^2/v_1,1} &= R_{\infty,1} + R_{1,1} \\
R_{\infty,0} + R_{\infty,1} &= -R_{\infty,\infty}
\end{aligned}
\tag{C.6}
$$

so that only three residues remain,

$$
(y - 1/y)^2 \mathcal{I}_{sc} = R_{1,1} - R_{\infty,\infty} - R_{0,0}.
\tag{C.7}
$$

Two of them are easily computed as follows:

$$
\begin{aligned}
R_{0,0} &= \text{Res}_{v_1=0}\text{Res}_{v_2=0}\frac{1}{v_1 v_2}\left(\frac{y-v_1/y}{v_1-1}\right)^{a_1}\left(\frac{y-v_2/y}{v_2-1}\right)^{a_2}\left(\frac{v_1 v_2-1}{y-v_1 v_2/y}\right)^{a_3} \\
&= (-y)^{a_1+a_2-a_3}.
\end{aligned}
\tag{C.8}
$$

$$
\begin{aligned}
R_{\infty,\infty} &= \text{Res}_{v_1=\infty}\text{Res}_{v_2=\infty}\frac{1}{v_1 v_2}\left(\frac{y-v_1/y}{v_1-1}\right)^{a_1}\left(\frac{y-v_2/y}{v_2-1}\right)^{a_2}\left(\frac{v_1 v_2-1}{y-v_1 v_2/y}\right)^{a_3} \\
&= \text{Res}_{\tilde{v}_1=0}\text{Res}_{\tilde{v}_2=0}\frac{1}{\tilde{v}_1 \tilde{v}_2}\left(\frac{\tilde{v}_1 y-1/y}{1-\tilde{v}_1}\right)^{a_1}\left(\frac{y\tilde{v}_2-1/y}{1-\tilde{v}_2}\right)^{a_2}\left(\frac{1-\tilde{v}_1\tilde{v}_2}{\tilde{v}_1\tilde{v}_2 y-1/y}\right)^{a_3} \\
&= (-y)^{-a_1-a_2+a_3}.
\end{aligned}
\tag{C.9}
$$

These two contributions sum up to the Coulomb index

$$
R_{\infty,\infty}+R_{0,0}=\mathcal{I}_{\text{reg}}=\frac{(-y)^{a_1+a_2-a_3}+(-y)^{a_3-a_1-a_2}}{(y-1/y)^2},
\tag{C.10}
$$

in the chamber $\mathcal{C}$ where $\zeta_1>0, \zeta_2>0, \zeta_3<0$. The last one, $R_{1,1}$ is recognized as the Jeffrey-Kirwan residue $\Omega_Q$ computing the full index in the same chamber. Therefore, the scaling index is equal to the sum of the single-centered index and the minimal modification part,

$$
\mathcal{I}_{\text{sc}}=\mathcal{I}-\mathcal{I}_{\text{reg}}=\Omega_S+H.
\tag{C.11}
$$

# D Generating series for scaling indices

In this section, we evaluate the generating series for the scaling part $\mathcal{I}_{\text{sc}}=\mathcal{I}_{\text{same}}+\mathcal{I}_{\text{uneq}}$ of the Witten index, for cyclic quivers with $K=3$ or $K=4$ nodes. Since the generating series of $Z_{\text{same}}$ was evaluated in §4.6 for arbitrary $K$, it remains to evaluate the generating series $Z_{\text{uneq}}$ in (4.52).

For $K=3$, the generating series can be evaluated using

$$
\sum_{a_1=1}^{\infty}\sum_{a_2=1}^{\infty}\sum_{a_3=1}^{\infty}\Theta(a_1+a_2-a_3)x_1^{a_1}x_2^{a_2}x_3^{a_3}=\frac{x_1 x_2 x_3(1+x_3-x_1 x_3-x_2 x_3)}{(1-x_1)(1-x_2)(1-x_1 x_3)(1-x_2 x_3)},
\tag{D.1}
$$

and suitable permutations thereof (where $\Theta(a)=1$ for $a\geq 0$ and $0$ for $a<0$). While the resulting expression for $Z_{\text{uneq}}$ is unilluminating, we find that the sum of equal and unequal sign contributions nicely combines into

$$
Z_{\text{same}}+Z_{\text{uneq}}=Z_S+Z_H,
\tag{D.2}
$$

where

$$
Z_S=\frac{x_1^2 x_2^2 x_3^2}{(1-x_1 x_2)(1-x_1 x_3)(1-x_2 x_3)(1-x_1 x_2-x_1 x_3-x_2 x_3-(y+y^{-1})x_1 x_2 x_3)}
\tag{D.3}
$$

is the generating series of single-centered invariants, a special case of (4.9), while

$$
\begin{aligned}
Z_H &= -\frac{2}{(y-y^{-1})^2}Z_{\text{even}}+\frac{y+y^{-1}}{(y-y^{-1})^2}Z_{\text{odd}}, \\
Z_{\text{even}} &= \frac{x_1 x_2 x_3(x_1+x_2+x_3-2x_1 x_2 x_3)}{(1-x_1 x_2)(1-x_1 x_3)(1-x_2 x_3)}, \\
Z_{\text{odd}} &= \frac{x_1 x_2 x_3}{(1-x_1 x_2)(1-x_1 x_3)(1-x_2 x_3)}.
\end{aligned}
\tag{D.4}
$$

Noting that the Taylor coefficients of $Z_{\text{even}}$ (respectively, $Z_{\text{odd}}$) are equal to one for monomials $x_1^{a_1} x_2^{b_1} x_3^{c_1}$ obeying the triangle inequalities with $a_1 + a_2 + a_3$ even (respectively odd), we see that $Z_H$ is the generating series of the minimal modification term for 3-node quivers [11]

$$H(\{\gamma_1, \gamma_2, \gamma_3\}) = \frac{1}{(y-1/y)^2} \begin{cases} -2, & a_1 + a_2 + a_3 \text{ even} \\ y + 1/y, & a_1 + a_2 + a_3 \text{ odd} \end{cases}. \tag{D.5}$$

Similarly, for cyclic quivers with 4 nodes, using

$$\sum_{a_1,\ldots,a_4=1}^{\infty} \Theta(a_2 + a_3 + a_4 - a_1) x_1^{a_1} \ldots x_4^{a_4} = \frac{x_1^3 x_2 x_3 x_4}{(1-x_1)(1-x_1 x_2)(1-x_1 x_3)(1-x_1 x_4)}$$

$$\sum_{a_1,\ldots,a_4=1}^{\infty} \Theta(a_1 + a_2 - a_3 - a_4) x_1^{a_1} \ldots x_4^{a_4}$$

$$= \frac{x_1 x_2 x_3 x_4 \left(1 + x_1^2 x_2 x_3 x_4 - x_1 x_2 (x_4 + x_3 (1 - x_2 x_4 - x_4))\right)}{(1-x_1)(1-x_1 x_3)(1-x_2 x_3)(1-x_1 x_4)(1-x_2 x_4)(1-x_3)(1-x_4)} \tag{D.6}$$

we find that the generating series for $Z_{\text{uneq}}$ nicely combines with $Z_{\text{same}}$ into $Z_S + Z_H$, where $Z_S$ is given by (4.9) while

$$Z_H = \frac{\tau_4 \left(\tau_3 - \tau_1 + (y + 1/y)(1 - \tau_4)\right)}{(y-1/y)^2 \prod_{1 \le i < j \le 4}(1 - x_i x_j)}. \tag{D.7}$$

The Taylor coefficients of $Z_H$ turn out to reproduce the minimal modification of the Coulomb index, for 4-node cyclic quivers, given in the chamber $\mathcal{C}$ for $a_2 + a_3 + a_4 > a_1 > a_2 > a_3 > a_4$ by [12, (4.13)]

$$g(\{\gamma_1, \ldots, \gamma_4\}) = \frac{(-1)^{1+a_1+\cdots+a_4}}{(y-1/y)^3} \tag{D.8}$$

$$\times \begin{cases} y^{a_1+a_2+a_3-a_4} - y^{a_1+a_2-a_3-a_4} - y^{a_1+a_3-a_2+a_4} - y^{a_2+a_3-a_1-a_4} - (y \to 1/y) \\ y^{a_1+a_2+a_3-a_4} - y^{a_1+a_2-a_3-a_4} - y^{a_1+a_3-a_2+a_4} - (y \to 1/y) \end{cases},$$

where the first and second lines correspond to $a_2 + a_3 > a_1 + a_4$ and $a_2 + a_3 < a_1 + a_4$, respectively. Indeed, applying the projection operator (2.20) we find

$$H(\{\gamma_1, \ldots, \gamma_4\}) = \frac{(-1)^{1+a_1+\cdots+a_4}}{(y-1/y)^3} \tag{D.9}$$

$$\times \begin{cases} -a_4(y^2 - 1/y^2), & a_2 + a_3 > a_1 + a_4, \ a_1 + \cdots + a_4 \text{ even} \\ -2a_4(y - 1/y), & a_2 + a_3 > a_1 + a_4, \ a_1 + \cdots + a_4 \text{ odd} \\ \frac{a_1-a_2-a_3-a_4}{2})(y^2 - 1/y^2), & a_2 + a_3 < a_1 + a_4, \ a_1 + \cdots + a_4 \text{ even} \\ (a_1 - a_2 - a_3 - a_4)(y - 1/y), & a_2 + a_3 < a_1 + a_4, \ a_1 + \cdots + a_4 \text{ odd} \end{cases}.$$

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
