# Peer review of "Multi-centered black holes, scaling solutions and pure-Higgs indices from localization"

_SciPost Physics, doi:SciPost Phys. 11, 023 (2021)_

## Round 2 · Referee Report · Anonymous (Referee 1) · 2021-6-20

Strengths

  1. Connects two different approaches for computing the refined index of QQM (the first is given by a Jeffrey-Kirwan residue formula and the second is an ad-hoc enumeration of colinear states satisfying an integrability condition a.k.a the Denef Equation) via a saddle point analysis.

  2. Identifies the source of `pure-Higgs' indices from the presence of additional saddles when the quivers have closed cycles.

  3. Results are shown to hold for quivers with an arbitrary number $K$ of nodes. Explicit details are given for $K=3$ and $K=4$.

  4. A mathematica package is provided in ref [33] for computing the formulas of interest in this paper. (Although the link is broken due to an ASCII problem with the "~" symbol in the link)

Weaknesses

None

Report

The paper in question provides a derivation of a counting formula proposed and checked in a series of papers by Manschot, Pioline and Sen (MPS) which gives the refined index of $\mathcal{N}=4$ quiver quantum mechanics (QQM). This is done by starting from the exact expression for the refined index of QQM in terms of a Jeffrey-Kirwan residue integral, and evaluating it by steepest descent. This provides a nice interpretation of the so-called `pure-Higgs' refined indices proposed by MPS.

This paper certainly meets the criteria for publication in Scipost.

Requested changes

None

---

## Round 2 · Referee Report · Anonymous (Referee 2) · 2021-6-28

Strengths

  1. The authors gave an explicit connection between the Coulomb phase Manschot-Pioline-Sen formula and the exact path integral by Hori-Kim-Yi.

  2. An alternate computation of the exact path integral is offered, which could prove useful down the road.

Weaknesses

none

Report

The manuscript establishes a long-sought-after connection between the exact path integral by Hori et.al. of 2014 for refined indices of gauge quantum mechanics and the general solution to the wall-crossing problem by Manschot et. al. of 2010. The authors achieved this by giving an alternate method for evaluating the former exact path integral by handling the auxiliary D integration differently.

The latter employs the Coulomb phase dynamics for the main tool and must be supplemented by additional input data on wall-crossing-safe sectors, while the former, resulting in a JK residue formula, computes everything in a single sweep. Although this may give an impression that the latter is outdated, this is not so because its deeper understanding of the wall-crossing pattern, if combined with the former, would give a more complete picture of the vacuum structure than refined indices alone.

In particular, this combination may lead us to a more systematic approach to the wall-crossing-safe part of the refined indices, which would be responsible for BPS black hole entropies if computed in the large rank limit. Although the same has been achieved example by example by combining the two existing sets of results, a direct evaluation of the wall-crossing-safe part of indices had remained out of reach. This new formulation brought us a significant step closer to this general goal and in fact gave a concrete routine for some of the simplest examples . One thing that remains unclear is whether this alternate form of the path integral evades the prohibitively heavy combinatorics that accompanied the JK residue formula for large-rank theories. Both clearly deserve further investigation.

---

## Editorial Decision

published